Manuscript prepared for Clim. Past
with version 2015/11/06 7.99 Copernicus papers of the LaTeX class copernicus.cls.
Date: 22 August 2016

# Testing the impact of stratigraphic uncertainty on spectral analyses of sedimentary series

Martinez Mathieu[1], Kotov Sergey[1], De Vleeschouwer David[1], Pas Damien[2], and Pälike Heiko[1]

[1]MARUM – Centrum for Marine Environmental Sciences, Leobenerstr., Universität Bremen, D-28359, Germany
[2]Pétrologie sédimentaire, B20, Géologie, Université de Liège, Sart Tilman, B-4000 Liège, Belgium
*Correspondence to:* Mathieu Martinez (mmartinez@marum.de)

**Abstract.** Spectral analysis is a key tool for identifying periodic patterns in sedimentary sequences, including astronomically related orbital signals. While most spectral analysis methods require equally-spaced samples, this condition is rarely achieved either in the field or when sampling sediment core. Here, we propose a method to assess the impact of the uncertainty or error made on the measurement of the sample stratigraphic position on the resulting power spectra. We apply a Monte-Carlo procedure to randomise the sample steps of depth series using a gamma distribution. Such a distribution preserves the stratigraphic order of samples, and allows controlling the average and the variance of the distribution of sample distances after randomisation. We apply the Monte-Carlo procedure on two geological datasets and find that gamma distribution of sample distances completely smooths the spectrum at high frequencies and decreases the power and significance levels of the spectral peaks in an important proportion of the spectrum. At 5% of stratigraphic uncertainty, a small portion of the spectrum is completely smoothed. Taking at least 3 samples per thinnest cycle of interest should allow this cycle to be still observed in the spectrum, while taking at least 4 samples per thinnest cycle of interest should allow its significance levels to be preserved in the spectrum. At 10 and 15% uncertainty, these thresholds increase, and taking at least 4 samples per thinnest cycle of interest should allow the targeted cycles to be still observed in the spectrum. In addition, taking at least 10 samples per thinnest cycle of interest should allow their significance levels to be preserved. For robust applications of the power spectrum in further studies, we suggest to provide a strong control of the measurement of the sample position. A density of 10 samples per putative precession cycle is a safe sampling density for preserving spectral power and significance level in the Milankovitch band. For lower sampling density, the use of gamma-law simulations should help in assessing the impact of stratigraphic uncertainty in the power spectrum in the Milankovitch band. Gamma-law simulations can also model the distortions of the Milankovitch record in sedimentary series due to variations in the sedimentation rate.

## 1 Introduction

Spectral analysis methods have become a key tool for identifying Milankovitch cycles in sedimentary series and are a crucial tool in the construction of robust astronomical time scales (Hinnov, 2013). The climatic or environmental proxy series that form the subject of spectral analyses are generally the result of measurements on rock samples collected from a sedimentary sequence, consisting of cores or outcrops. Most of spectral analysis methods (Fourier Transforms and derivatives, such as the Multi-Taper Method) require equally-spaced depth- or time-series, which implies that samples need to be taken at a constant sample step (Fig. 1a). Unfortunately, this is rarely achieved, especially for sedimentary sequences sampled in outcrops (e.g., Figs. 1b-c and e). Often, an uncertainty of ~5-15% is observed in the thickness or distance measurements, even when using a Jacob's staff (Weedon and Jenkyns, 1999). In core sediments, uncertainties in the sample position are also observed when performing physical sampling at very high resolution or because of core expansion phenomena (Hagelberg et al., 1995) or imperfect coring (Ruddiman et al., 1987).

Although uncertainties exist on the actual position of samples, few case studies document their effect on the identification of periodic patterns. Moore and Thomson (1991) recognised that perturbations of the regular sampling scheme (i.e. jittered sampling) impact the power spectrum by reducing spectral power in the high frequencies. Huybers and Wunsch (2004) and Martinez and Dera (2015) address an analogous problem by assessing the effect of sampling uncertainty on the age model of a calibrated time series that is plotted against numerical age. However, none of these studies explicitly addresses the impact of errors in the measurement of the sample position on uncertainties in the power spectrum amplitudes. In this study, we address this problem by quantifying the impact of such errors on the frequency and power distribution. Therefore, we provide a new procedure that is based on a Monte-Carlo approach for randomising the distance between two successive samples in a sedimentary series. The resulting simulated series are subsequently used to assess the impact of the sample-position error on spectral analyses. We first apply the procedure to a theoretical example, and then to two previously published geological datasets, one as-regularly-as-possible sampled and another irregularly sampled.

## 2 The error model

In this paper, the term "stratigraphic uncertainty" refers to the uncertainty of the sample positions. Testing the impact of the stratigraphic uncertainty on the spectral analyses requires a randomisation procedure that reflects typical errors made during measurements of the stratigraphic position of samples. Figures 1c to 1e illustrate the consequences of the stratigraphic uncertainties on a geological series (here the La Charce series, see section 3.1). Fig. 1c compares the real sampling made on this series (in red) to an ideal sampling in which samples are taken at a strictly even sample distance (in black). Errors in the sample positions distort the sedimentary series: some intervals are compressed

while some others are dilated. Ideally, all sample distances should be strictly the same, so that the distribution of sample distances should be concentrated on only one value (Fig. 1d). In reality, as uncertainties exist on the sample positions, the sample distances show a distribution over a certain range of values, which depends on the accuracy with which the distance measurements have been taken (Fig. 1e). In the case of the La Charce series, the standard deviation of the sample distances is

assessed at 12.5% of the average sample distance (the method to estimate this standard deviation is provided in section 4). If the error in the distance measurement was systematic, one should expect the same level of error in the total length of the series. However in total, the difference of the length of the series between the ideal case (all sample distances strictly the same) and the real case is only 1.4% of the total length of the series (Fig. 1c). Each sample distance is measured independently

from the other sample distances, so that each measurement can overestimate or underestimate the real distance between two successive samples. The errors thus compensate each others, implying that the process at the origin of the error measurements is not systematic but random.

Three conditions must be respected to design the error model: (i) the stratigraphic order of samples is hard set and must not be changed by the randomisation process (e.g., Fig. 1c), (ii) the average

and standard deviation of sample steps must be maintained during the randomisation process, (iii) the error model must be random. These conditions can be achieved if the error model randomises the sample distances rather than the sample positions. In that case, the probability density function should have a positive and continuous distribution (i.e. values obtained after randomisation are continuous and positive). In addition, the average sample step and the standard deviation of the distance

between two successive samples are known and should be parameterized.

The gamma distribution fulfils all these conditions. The gamma distribution is continuous and has a positive support. Parameters $k$ and $\Theta$ respectively define the shape of the distribution and its range of values. The mean ($E$) of the density of probability is defined as (Burgin, 1975):

$$E = k * \Theta \tag{1}$$

and its variance ($\sigma^2$)

$$\sigma^2 = k * \Theta^2 = E * \Theta \tag{2}$$

Both the mean ($E$) and the variance ($\sigma^2$) are known, as they correspond to the mean and variance of the sample steps, and they can be quantified in the field (see Section 4 for a discussion on the variance of sample steps). Therefore, $k$ and $\Theta$ can be parameterized using the following relations:

$$\Theta = \frac{\sigma^2}{E} \tag{3}$$

$$k = \frac{E}{\Theta} \tag{4}$$

Various gamma probability density functions are shown in Fig. 2. A high variance-to-mean ratio corresponds to a high $\Theta$-parameter value compared to the $k$-parameter. The resulting density probability function corresponds to an exponential probability function in the most severe and spectrum-destructive case. This distribution corresponds to sampling conditions during which no control was exerted on the stratigraphic position of samples, so that the uncertainty on the sample position is at a maximum. Obviously, this situation is not a realistic case to reflect geological practice.

In the opposite case, a low variance-to-mean ratio corresponds to a low $\Theta$-parameter value compared to the value of the $k$-parameter. The resulting density probability function is close to a Gaussian curve, although bound on one side to 0, so that the curve has a positive support. This case corresponds to geological sampling during which the position of each sample was carefully measured and reported with respect to the stratigraphic column. Nevertheless, even in this case, stratigraphic uncertainties are unavoidable, mainly because of outcrop or core conditions. Interestingly, this latter case has a similar distribution to the distribution of sample distances in the La Charce series (Fig. 1e). This illustrates that the gamma model is well adapted for simulating the errors made on the measurement of the sample distances.

## 3    The geological datasets

Two published geological datasets were used here to assess the effect of stratigraphic uncertainty on power spectra.

### 3.1    Gamma-ray spectrometry from La Charce (Valanginian, Early Cretaceous)

A total of 555 gamma-ray spectrometry measurements were performed *in situ* on the La Charce section (Department of Drôme, SE France; Martinez et al., 2013, 2015). The section is composed of marl-limestone alternations that were deposited in a hemipelagic environment during the Valanginian and Hauterivian stages (~134-132 Ma, Early Cretaceous; Martinez et al., 2015). Detailed analyses of the clay mineralogical, geochemical, and faunal contents indicated that these alternations reflect orbital climate forcing. Gamma-ray spectrometry measurements were used to identify the precession, obliquity and 405-kyr eccentricity cycles (see  Martinez et al., 2015).

Gamma-ray spectrometry measurements were performed directly in the field with an as regular as possible sample step of 0.20 m. Before each measurement, rock surfaces were first cleaned to remove reworked material and flattened to prevent any border effects that could affect the measurement value. Each measurement was performed using a SatisGeo GS-512 spectrometer, with a constant acquisition time of 60 seconds (more details are provided in  Martinez et al., 2013).

### 3.2 Magnetic susceptibility from La Thure section (Givetian, Middle Devonian)

The second case study consists of the 184-m-thick continuous early-Givetian to early-Frasnian se-
quence of the La Thure section (~383-380 Ma, Middle-Late Devonian; De Vleeschouwer and Par-
nell, 2014; De Vleeschouwer et al., 2015; Pas et al., 2016). The Givetian sequence is composed of
bedded limestone, mainly deposited in a shallow-water rimmed-shelf characterised by a large set
of internal and external rimmed-shelf environments (Pas et al., 2016). The overlying early Fras-
130 nian sequence is dominated by shale deposited in a siliciclastic drowned platform (Pas et al., 2015).
The magnetic susceptibility (MS) data from the La Thure section, in combination with three other
MS datasets from the Dinant Syncline in southern Belgium and northern France were used by
De Vleeschouwer et al. (2015) to make an estimate of the duration of the Givetian Stage, and subse-
quently to calibrate the Devonian time scale (De Vleeschouwer and Parnell, 2014). Spectral analysis
of the MS data from the La Thure section revealed the imprint of different Milankovitch astronomi-
cal parameters, including eccentricity, obliquity and precession (Fig. 3c in De Vleeschouwer et al.,
2015). A total of 484 samples were taken along the 184-m thick sequence, with an irregular sample
step that varied between 20-45 cm, depending on outcrop conditions (average sample step: 38 cm).
Magnetic susceptibility measurements were performed using a KLY-3S instrument (AGICO, noise
level $2 * 10^{-8} SI$) at the University of Liège (Belgium) (more details provided in De Vleeschouwer
et al., 2015).

### 4 Implementation of the models in the stratigraphic-uncertainty tests

Weedon and Jenkyns (1999) estimated the error on the stratigraphic position of a sample as 5.3%,
by measuring the thickness of the same sequence twice. The La Charce section, one of the datasets
treated here, has been measured multiple times in different publications. The thickness of the studied
section was assessed at 106 m, 109 m and 116 m (Bulot et al., 1992; Martinez et al., 2013; Reboulet
and Atrops, 1999) with an average of $110.3 \pm 5.1$ m, which represents a relative uncertainty of 4.6%
in the total thickness of the series. In the field, the distance between two successive samples was
measured independently from the construction of the log, providing an independent assessment of
150 the distribution of the actual distance between two successive samples. The average sample step is
20 cm, with a standard deviation of the sample steps of 2.5 cm, which corresponds to an uncertainty
of 12.5% in the average sample step (Fig. 1e).

Based on the assessments summarised in the previous paragraph, we tested three different levels
for the error on the measurement of sample steps (5%, 10% and 15%), which we consider realistic
scenarios for geological sampling during fieldwork. We applied our Monte-Carlo based procedure
for randomising sample steps to a sinusoidal series, as well as to the two previously published ge-
ologic datasets described in section 3 (De Vleeschouwer et al., 2015; Martinez et al., 2013, 2015),
with three different error levels. During every Monte-Carlo simulation, the distance between two

points is randomised according to a gamma distribution, of which the mean corresponds to the distance between two points measured in the field, and of which the standard deviation corresponds to 5%, 10% or 15% of the measured distance. Each test consists of 1000 Monte-Carlo simulations, leading to 1000 different time series, each with a different distortion of the stratigraphic positions of samples.

Spectral analyses were performed using the Multi-Taper Method (MTM; Thomson, 1982, 1990), using three $2\pi$-tapers ($2\pi$-MTM analysis) and with the Lomb-Scargle method (Lomb, 1976; Scargle, 1982). For the $2\pi$-MTM analysis, confidence levels of the spectra of the original geological datasets tested were calculated using the Mann and Lees (1996) approach (ML96), with median-smoothing calculated with the method of the Tukey's end point rule, as suggested by Meyers (2014). The window width for the median-smoothing was fixed at 20% of the Nyquist Frequency (the highest frequency which can be detected in a time series), as evaluated empirically by Mann and Lees (1996). MTM analysis requires strictly regular sample steps to be performed, so that geological datasets were linearly interpolated at the average sample distance of the original series before and after randomisation. We limit the loss of amplitude in the high-frequency fluctuations due to resampling by applying an optimized procedure to find the best starting point of the interpolated series. To our knowledge, this procedure is new, and we therefore describe it in Appendix 1. We provide the corresponding R-function in the supplementary material. The sum of sinusoid series is generated with a regular sample step of 1 arbitrary unit. After randomisation, the depth-randomised series were linearly interpolated at 1 arbitrary unit.

Lomb-Scargle spectra were calculated with the REDFIT algorithm (Schulz and Mudelsee, 2002) available in the R-package dplR (Bunn, 2008, 2010; Bunn et al., 2015). The Lomb-Scargle method calculates the spectrum of unevenly-sampled series. Lomb-Scargle power spectra can be biased in the high frequencies due to the non-independency of the frequencies (Lomb, 1976; Scargle, 1982); however, the REDFIT algorithm corrects the power spectrum by fitting a red-noise model to the spectrum (Mudelsee, 2002; Schulz and Mudelsee, 2002). Here, we applied no segmentation to the series and a rectangular window. This parameterization maximises the effect of sample step randomisation on the spectrum.

During each test, both MTM and REDFIT Lomb-Scargle power spectra were calculated for each of the 1000 Monte-Carlo distorted series. Subsequently, the average power spectra and the range of powers covered by 95% of the simulations were calculated for the MTM and Lomb-Scargle analyses. The confidence levels of the datasets deduced from the red-noise fit of the spectral background were calculated after each simulation. The average power of the confidence levels and the range of powers of the confidence levels covered by 95% of the simulations were calculated and directly plotted ontop of the simulated spectra. The sum of sinusoids series does not need correction for red noise and the raw Lomb-Scargle spectra are shown. The two geological datasets show a red-noise background and the REDFIT-corrected Lomb-Scargle spectra were shown.

We finally provide a quantification of the relative change in spectral power, using the following criterion:

$$E_r(f) = \left| \frac{P_{ori}(f) - P_{ave}(f)}{P_{ave}(f)} \right| \tag{5}$$

with $f$: the frequencies explored in the spectral analyses, $E_r$: the relative change of power, $P_{ori}(f)$:
the power spectrum before randomization at frequency $f$, and $P_{ave}(f)$: the average power spectrum
of the 1000 simulations at frequency $f$.

## 5   Application to a sum of sinusoids

The effect of randomising the sample positions within the section is first tested on a sum of pure
sinusoids. A dataset of 600 points is generated with a sample step of 1 arbitrary unit. The series is
a sum of 24 sinusoids, having equal amplitudes and different frequencies: frequencies range from
0.02 to 0.48 cycles/arbitrary unit and increase with increments of 0.02 cycles/arbitrary unit (Figs.
3a, b). Fig. 3 shows the $2\pi$-MTM and Lomb-Scargle spectra of the sum of sinusoids before and
after applying 1000 Monte-Carlo simulations of distorted sample distances. The grey zones indicate
the interval covering 95% of the power in the 1000 simulations. The average spectrum of these
simulations is shown in orange for the test with 5% stratigraphic uncertainty (Figs. 3c, d), red for
10% uncertainty (Figs. 3e, f), and brown for 15% uncertainty (Figs. 3g, h). The most striking feature
after gamma-model randomisation is the progressive and strong decrease of the power spectrum
towards the high frequencies, even when the lowest level of uncertainty (5%) is considered.

Fig. 4 notably shows the relative change in power of the average spectrum after applying the 1000
simulations. At 5% uncertainty, a decrease of 50% in the power spectrum is observed in the $2\pi$-MTM
spectrum at 57% of the Nyquist frequency, equivalent to 3.5x the average sample distance. The level
of 50% of decrease in the power spectrum is rather observed in the Lomb-Scargle spectrum at 80%
of the Nyquist frequency, i.e. 2.5x the average sample distance. This implies that even for a very low
level of noise, the values of the power spectrum can be largely underestimated in the upper half of
the spectrum. At 10% uncertainty, a decrease of 50% in the power spectrum is observed at 38-39%
of the Nyquist frequency, both in the Lomb-Scargle and the $2\pi$-MTM spectra, which is equivalent
to 5.2x the average sample distance. Finally, at 15% uncertainty, both Lomb-Scargle and $2\pi$-MTM
indicate that 50% of decrease in the power spectrum has occurred at 27% of the Nyquist frequency,
which is equivalent to 7.4x the average sample distance. This example shows that the worse the
control of the sample position in the sedimentary series is, the more samples per cycle one needs to
limit the loss of power of the cycles targeted.

Stratigraphic uncertainty does not only trigger loss of power of the spectral peaks, it also increases
the power spectral background (Fig. 3). At 5% and 10% uncertainties, the average and background
spectrum still preserve the structure of individual peaks in both $2\pi$-MTM and Lomb-Scargle analyses
(Figs. 3c-f). Indeed, spectra for individual Monte-Carlo simulations still exhibit spectral peaks at

these frequencies although they are characterised by variable power and deviations in the frequencies at which the peaks are localised. However, at 15% uncertainty, the average power at the highest frequencies is flattened and the structure of the peaks is not distinguishable anymore (Figs. 3e-h). This zone of the spectrum cannot be regarded as reliably interpretable. These analyses from a sum of pure sinusoids show that the higher the stratigraphic uncertainty is, the higher is the loss in power of the spectral peaks and the more the low frequencies are affected by this loss of power. At 15% uncertainty, the spectrum is flattened in the highest frequency and cannot be interpreted in this part of the spectrum. Because of its higher frequency resolution, the Lomb-Scargle analysis displays higher spectrum background levels than the $2\pi$-MTM analysis. It however changes very little the highest frequency that can be interpreted, even at 15% uncertainty.

It should be noticed that in the case of pure sinusoids, the signal is only composed of pure harmonics concentrating the spectral power at specific frequencies. This implies that a small shift in the sample position triggers a strong decrease of the average power spectrum at these specific frequencies. In addition, in this theoretical example, the sample distance before the randomisation procedure was strictly constant (1 arbitrary unit). More realistically, spectra of geological datasets are rather composed of a mixture of harmonics, narrow-band and background components, and the sample distances are not strictly constant. For instance, because of variations in the sedimentation rates, the sedimentary expression of the orbital cycles is not focused on specific frequencies but rather expressed on ranges of frequencies (e.g. Weedon, 2003, p. 132). This can add some noise in the high frequencies, and blur the spectra even more than in the case of pure sinusoids. In the following, the results of the application of the test on two geological datasets are shown.

## 6 Application to geological datasets

### 6.1 Spectral analysis prior to randomisation

#### 6.1.1 The La Charce series

Prior to performing $2\pi$-MTM analyses, the gamma-ray series was detrended using a best-fit linear regression, linearly interpolated to 0.20 m sample distance, and standardised to zero average and unit variance (Fig. 5). Prior to REDFIT Lomb-Scargle analysis, the datasets (raw and randomised) were simply linearly detrended using a best-fit linear regression and standardised.

The $2\pi$-MTM analysis of the La Charce section shows two main significant bands (>99% Confidence Level, hereafter abbreviated CL) at 20 m and from 1.3 to 0.8 m (Fig. 6a). The peak at a period of 20 m has been interpreted as the imprint of 405-kyr eccentricity forcing, while the group of peaks at 1.3 to 0.8 m has been dominantly related to precession (Boulila et al., 2015; Martinez et al., 2013, 2015). The REDFIT spectrum shows two bands of periods exceeding the 99% CL at 18 m and from 1.4 to 0.8 m (Fig. 6b). These periods are similar to the periods observed in the $2\pi$-MTM spectrum.

The small differences in periodicity observed in the lowest frequencies are likely to be related to the difference in the frequency resolution between the two methods. In addition, the REDFIT spectrum as parameterised here produces narrower peaks than the multi-taper spectrum, so that the lowest frequencies in the REDFIT spectrum are composed of a group of narrow peaks, rather than a single broad peak observed in the $2\pi$-MTM spectrum.

The autoregressive coefficient, a measure for the redness of the spectrum, is assessed at 0.440 in the $2\pi$-MTM analysis, while it is assessed at 0.468 in the REDFIT analysis (Table 1). The S0-value, the average power of the red-noise process within the entire spectrum, is $3.54 \cdot 10^{-4}$ in the MTM analysis, while it is 0.398 in the REDFIT analysis (Table 1). This difference in the S0-value is due to the difference of signal treatment when calculating the MTM or the REDFIT spectrum.

### 275 6.1.2 The La Thure series

Prior to performing $2\pi$-MTM analyses, the magnetic susceptibility series was detrended by subtracting a piecewise best-fit linear regression (Fig. 7a). The series was then linearly interpolated to a sample distance of 0.38 m, and the trend of the variance was removed by dividing the series by its instantaneous amplitude smoothed with a LOWESS regression with a 10% coefficient (Fig. 7b).
Such an approach allows the series to have a stationary mean and variance (Fig. 7c). The series was subsequently standardised (average = 0; standard deviation = 1). Prior to the REDFIT analysis, the identical procedure was applied, except for interpolation at an even sample step, as this is not required by the Lomb-Scargle method.

The $2\pi$-MTM analysis of the La Thure section shows significant periods at 39 m (>99% CL)
interpreted as the manifestation of the 405-kyr eccentricity cycle (De Vleeschouwer et al., 2015), at 7.8 m (>95% CL) interpreted as 100-kyr eccentricity cycles, a group of significant periods from 2.8 m to 2.2 m (99% CL) interpreted as obliquity, and a group of significant periods from 1.6 to 1.1 m (>95% and >99% CL) interpreted as precession (Fig. 6c). In the lowest frequencies, the REDFIT spectrum (Fig. 4f) shows a group of peaks centred on 30-40 m (>99% CL), a peak at 13 m (>95%
CL), which is not significant in the $2\pi$-MTM spectrum. Conversely, the period at 7.9 m observed in the $2\pi$-MTM spectrum does not reach the 90% CL in the REDFIT spectrum. These differences are likely related to the difference in frequency resolution between both methods, and to the fact that REDFIT spectra as parameterised here produce narrower peaks than the $2\pi$-MTM spectra. In the REDFIT spectrum, the obliquity band shows two periods at 3.3 m (95% CL), and 2.3 m (>95% CL).
The precession band shows periods at 1.5 m (>90% CL), 1.1 m (>99% CL) and at 0.9 m (>95% CL).

The autoregressive coefficient of the red-noise background level is assessed at 0.657 in the $2\pi$-MTM analysis, and at 0.407 in the REDFIT analysis (Table 1). The difference in the autoregressive coefficient is due to the method of calculation of the red-noise background (from the spectrum in the MTM analysis, from the time series in the REDFIT analysis; Mann and Lees, 1996; Meyers,
2014; Mudelsee, 2002). The Lomb-Scargle analysis also tends to produce higher powers in the high

frequencies, thus reducing the autoregressive coefficient estimate in the REDFIT analysis (Schulz and Mudelsee, 2002). This difference also illustrates the difficulty in calculating the autoregressive coefficient when the redness of the spectrum increases (see Meyers, 2012). Finally, the S0-value is assessed at $1.67 \cdot 10^{-3}$ in the $2\pi$-MTM analysis, and at 0.890 in the REDFIT analysis (Table 1).

## 6.2 Impact on the power spectrum of randomising the sample distances

### 6.2.1 The La Charce series

At 5% uncertainty, the average $2\pi$-MTM spectrum of the La Charce still shows periods at 20.5 m as well as several periods around 1 m exceeding the 99% CL (Fig. 8a). At 10% uncertainty, the peak at 0.8 m does not exceed the 95% CL (Fig. 8b), and it is completely smoothed at 15% uncertainty (Fig. 8c). The increasing level of stratigraphic uncertainty progressively smooths the average spectrum, with the highest frequencies most affected (Figs. 8d-f). Notably at 5% uncertainty, fluctuations of the spectrum at frequencies higher than 81% of the Nyquist frequency are suppressed (Table 2). At 10% and 15% uncertainty, this threshold decreases to respectively 58 and 43% of the Nyquist frequency (Figs. 8d-f). Increasing levels of uncertainty also tend to reduce the power of the spectral peaks in an increasing portion of the spectrum. At 5% uncertainty, the average spectrum of the simulations is practically identical to the spectrum of the original series from frequency 0 to 27% of the Nyquist frequency (Fig. 8d). This range is reduced to 0 - 19% of the Nyquist frequency at 10% uncertainty (Fig. 8e) and to 0 - 18% of the Nyquist frequency at 15% uncertainty (Fig. 8f).

In the REDFIT spectrum with 5% of stratigraphic uncertainty, the periods at 20.5 m and around 1 m still exceed the 99% CL (Fig. 9a). Like in the $2\pi$-MTM analyses, the period at 0.8 m does not exceed the 99% CL at 10% uncertainty, while it is completely smoothed at 15% uncertainty (Figs. 9b-c). The tendency of the Lomb-Scargle analysis to produce high-power peaks in the high frequencies limits the effect of the smoothing of the spectrum at 5% uncertainty (Fig. 9d). However, at 10 and 15% uncertainties, fluctuations in the spectrum at frequencies higher than respectively 58 and 42% of the Nyquist frequency are completely smoothed (Figs. 9e-f; Table 2). At 5% uncertainty, the average spectrum of the simulations cannot be distinguished from the spectrum of the original series from frequency 0 to 29% of the Nyquist frequency (Fig. 9d), while at 10 and 15% uncertainties, this range is restricted to 0 - 19% of the Nyquist frequency (Figs. 9e-f).

The average autoregressive coefficients of the 1000 simulations (with $\pm$ the interval covering 95% of the simulations) are assessed for 5, 10, and 15% of stratigraphic uncertainties at $0.433 \pm 0.025$, $0.432 \pm 0.037$, $0.434 \pm 0.048$, respectively, in the $2\pi$-MTM analyses, and at $0.468 \pm 0.002$, $0.467 \pm 0.003$, $0.467 \pm 0.006$, respectively, in the REDFIT analyses (Table 1). The average S0-values of the 1000 simulations are assessed for 5, 10, and 15% of stratigraphic uncertainties at $3.55 \cdot 10^{-4} \pm 0.13 \cdot 10^{-4}$, $3.58 \cdot 10^{-4} \pm 0.20 \cdot 10^{-4}$, $3.61 \cdot 10^{-4} \pm 0.25 \cdot 10^{-4}$, respectively, in the $2\pi$-

MTM analyses, and at $0.399 \pm 0.003$, $0.402 \pm 0.005$, $0.407 \pm 0.008$, respectively, in the REDFIT
analyses.

### 6.2.2   The La Thure series

At 5% uncertainty, the $2\pi$-MTM spectrum of the La Thure series still exhibits significant frequencies
at 39 m, 1.5 m and 1.1 m exceeding the 99% CL, and at 7.5 m, 2.9 m, 2.2 m and 1.6 m exceeding
the 95% CL (Fig. 10a). At 10% uncertainty, the 1.1-m peak is much smoother, centred on a period
of 1.2 m and only exceeds the 95% CL (Fig. 10b). The other periods of the precession, at 1.5 and
1.6 m, only exceed the 90 and 95% CL, respectively. The significant periods of the obliquity bands,
at 2.2 and 2.9 m, show weaker powers than in the spectrum of the original series, but still exceed
the 95% CL. At 15% uncertainty, the band of periods at 1.2 m is nearly entirely flattened and hardly
distinguishable from the spectral background (Fig. 10c). In addition, all frequencies from the obliq-
uity and the precession do not exceed the 95% CL. The reduction in the significance levels in the
precession and obliquity bands is the consequence of increasing loss in power of the spectral peaks at
high frequencies. At 5% uncertainty, the average spectrum of the simulations is practically identical
to the spectrum of the original series from frequency 0 to 52% of the Nyquist frequency (Fig. 10d),
while at 10 and 15% uncertainties, this range is restricted to 0 - 20% of the Nyquist frequency (Figs.
10e-f; Table 2).

    At 5% uncertainty, the REDFIT analysis still displays significant periods at 30-40 m exceeding the
99% CL, and a period at 2.3 m exceeding the 95% CL (Fig. 11a). The peak at 1.5 m does not exceed
anymore the 90% CL, while the peaks at 1.1 m and 0.9 m do not exceed anymore the 95% CL. At
10% uncertainty and 15% uncertainties, spectral peaks in the precession and the obliquity bands do
not reach the 95% CL anymore. The tendency of the Lomb-Scargle analysis to produce high-power
peaks in the high frequencies prevents strong smoothing of the power spectrum at 5% uncertainty.
However, at 10 and 15% uncertainties, all fluctuations of the power spectrum at frequencies higher
than 53% Nyquist frequency are flattened and not distinguishable (Table 2). The significance level in
the eccentricity band is still preserved in the average spectrum. At 10 and 15% uncertainty, the power
spectrum displays spectral peaks with reduced powers compared to the spectrum of the original
series, which impacts the significance levels at the obliquity and precession bands (Figs. 11d-f).
At 5% uncertainty the REDFIT spectrum of the La Thure series remains practically unchanged
compared to the spectrum of the original series from 0 to 58% Nyquist frequency (Fig. 11d), while
at 10 and 15% uncertainty this range is respectively restricted to 0 - 22% and 0 - 19% Nyquist
frequency (Figs. 11e-f).

    The average autoregressive coefficients of the 1000 simulations are assessed for 5, 10, and 15% of
stratigraphic uncertainties at $0.658 \pm 0.025$, $0.653 \pm 0.029$, $0.651 \pm 0.033$, respectively, in the $2\pi$-
MTM analyses, and at $0.406 \pm 0.004$, $0.405 \pm 0.008$, $0.404 \pm 0.013$, respectively, in the REDFIT
analyses (Table 1). The average S0-values of the 1000 simulations are assessed for 5, 10, and 15% of

stratigraphic uncertainties at $1.67 \cdot 10^{-3} \pm 0.04 \cdot 10^{-3}$, $1.67 \cdot 10^{-3} \pm 0.05 \cdot 10^{-3}$, $1.68 \cdot 10^{-3} \pm 0.07 \cdot 10^{-3}$, respectively, in the $2\pi$-MTM analyses, and at $0.894 \pm 0.011$, $0.900 \pm 0.019$, $0.904 \pm 0.008$, respectively, in the REDFIT analyses.

## 7 Discussion

### 7.1 Comparison of the results between the two geological datasets

In the $2\pi$-MTM simulations, the spectral peaks tend to be smoothed at 5% of stratigraphic uncertainty from ~80% Nyquist frequency to the Nyquist frequency, which implies that taking at least 3 samples per cycle of interest should not smooth the spectral peaks in the frequency band targeted (e.g., the Milankovitch cycles) (Table 2). In the REDFIT simulations, the tendency of the spectrum to produce high-power spectra in high frequencies even makes all the spectral peaks of the original spectrum still identifiable at 5% uncertainty. If a low level of stratigraphic uncertainty is maintained, practically all spectral peaks at frequencies below 80% Nyquist frequencies will be preserved. These thresholds dramatically decrease to 53% to 66% of Nyquist frequencies at 10% of stratigraphic uncertainty in all simulations, while it decreases to 42% to 53% of Nyquist frequency at 15% uncertainty. Thus, a medium level of stratigraphic uncertainty implies taking at least 4 samples per cycles of interest, while a high level of uncertainty implies taking at least 5 samples per cycle of interest.

Comparisons between original and average simulated spectra show that at 5% uncertainty, both are practically identical from 0 to 27% of Nyquist frequency in the La Charce series and from 0 to 52% of Nyquist frequency in the La Thure series. At 10 and 15% uncertainties, these ranges dramatically shift from 0 to 20-22% Nyquist frequency. Although differences exist in the variance of the average spectrum and in the frequency resolution between the $2\pi$-MTM and the REDFIT analyses, both analyses show, for each series, the same range of frequencies in which simulated and original spectra are identical. These thresholds imply that taking 4-8 samples per cycle of interest should limit loss of power of the spectral peaks in the targeted bands at 5% uncertainty. At 10 and 15% uncertainty, taking at least 10 samples per cycle of interest should limit the loss of power in the targeted band. Limiting the loss of power in the frequencies of interest appears to be crucial because the average power of the confidence levels remain unchanged after applying the simulations. Simulations of distortions of the geological series smoothes the spectrum by distributing the power spectrum from the spectral peaks to the surrounding frequencies. The calculation of confidence levels in the MTM analyses is based on a moving median of the power spectrum performed over a broad range of frequencies (usually 1/5 of the total spectrum; Mann and Lees, 1996). Thus, when distorting the time series, the distribution of the power spectrum over a narrow range of frequencies does not change the overall median of the power spectrum calculated over 1/5 of the total spectrum, and thus does not change the average level of confidence levels after simulations. The effect of time-series distortions on the power of confidence levels is even smaller in the REDFIT analysis, in which the confidence

levels are directly calculated on the time series itself and not on the spectrum (Mudelsee, 2002). The decrease of the power of the spectral peaks due to distortions of the geological series thus implies a decrease in the significance levels of the main cycles of the series. In case of low level of red noise, like in the La Charce series (Figs. 8-9), spectral smoothing and decrease in power in the precession 410 band does not strongly impact the interpretations, since the significance level in the precession band still exceed the 99% CL, even after implementation of a level of 15% of stratigraphic uncertainty. However, in case of strong red noise, like in the La Thure series, the decrease of power in high frequencies has a strong impact on the significance levels after implementation of the simulations. At a medium level of stratigraphic uncertainty (10%), taking 10 samples per cycle of interest is 415 needed to limit the loss of power in the cycles of interest and thus to limit the decrease in the level of significance of these targeted cycles.

As an example, if the targeted range of frequencies are the Milankovitch cycles, the shortest period of interest are the precession cycles. A density of 1 sample per 4 kyr should allow the detection of the spectral peaks in the precession band. A density of sampling of 1 sample per 2 kyr should then 420 ensure the detection of significant peaks in the precession band, even in case of strong red noise and medium-to-high levels of stratigraphic uncertainty. The minimum density of sampling being dependant on the level of red noise and stratigraphic uncertainty, we strongly recommend to apply the simulations developed here to assess the impact of stratigraphic uncertainty to the identification of significant spectral peaks in the sedimentary record.

425 **7.2    In which case to apply this test?**

Uncertainties in the measurement of sample position can practically not be avoided in outcrop conditions. The similarity between the topographic slope and the sedimentary dip, the absence or scarcity of marker beds, or the need to move laterally in a section can trigger disturbances in the sampling regularity. In core sedimentary sequences, non-destructive automated measurements such as X-ray 430 fluorescence, gamma-ray spectrometry or magnetic susceptibility should limit errors in the sample position. However, physical samplings (e.g. for geochemistry or mineralogy) are subject to small uncertainties, especially when the sampling resolution is very high. Core sedimentary series can in addition be affected by expansion of sediment caused by release of gas or release of overburden pressure (Hagelberg et al., 1995). This test is thus useful for geologists who wish to run spectral analyses 435 on sedimentary depth-series generated from outcropping sections or core samples. All analyses in this paper show that with higher uncertainty on the sample step, the low frequencies are increasingly affected. The relative change in power between the various tests all showed different patterns, and no general model could be deduced. The relative change in power at a given frequency depends on the dispersion of the sample step, on the method of spectral analysis, but also on the original 440 sedimentary sequence studied. Each depth-series generated from this sampling can be seen as one of the 1000 random simulations. The test randomises the sample position from the original series,

and produces a smooth version of the spectrum of the raw series. The generation of the raw series impacts on the test at frequencies having low powers (a small change in a weak power can trigger high values of relative change in power), and at high frequencies. The relative change in power does not depend on the size of the sample step itself, as the same proportion of the spectrum is affected for a given level of uncertainty. However, a control on the dispersion of the sample steps and the application of the test proposed here are needed to assess the dispersion of the sample distances during the sampling procedure and the impact of this dispersion on the spectrum. The question is how to assess the dispersion of the sample step in the field? If the section is well bedded, we suggest applying the same procedure as we did for La Charce, i.e. sample position measured independently from the bed thickness measurements, and precise reporting of the sample positions in the sedimentary log of the series. Orbital forcing can also be detected in a monotonous thick marly section, showing no apparent bedding (e.g., Ghirardi et al., 2014; Matys Grygar et al., 2014). In that case, we rather suggest measuring the total thickness of the sequence several times to assess the potential dispersion of the sample steps.

### 7.3  Implications for astronomical time scale and palaeoclimate reconstructions

Linking sedimentary cycles to orbital cycles or assessing the quality of an orbital tuning procedure often requires a good matching between the sedimentary period ratios and the orbital period ratios (Huang et al., 1993; Martinez et al., 2012; Meyers and Sageman, 2007) and/or the determination of the amplitude modulation of the orbital cycles (Meyers, 2015; Moiroud et al., 2012; Zeeden et al., 2015). On average, stratigraphic uncertainties trigger a decrease of the power spectrum of the main significant frequencies while distributing the power spectrum to the surrounding frequencies. In the studied geological data, stratigraphic uncertainties mostly impact the precession band, by decreasing the power and significance levels of the spectral peaks and multiplying the main frequencies for each individual runs. The occurrence of low-power spectral peaks in the precession bands, and the fact that frequency ratios between the precession and lower frequencies do not match the orbital frequency ratios are quite common in the geological data (e.g. Ghirardi et al., 2014; Huang et al., 2010; Thibault et al., 2016), and can be a consequence of stratigraphic uncertainties. Variations in the sedimentation rate produces a similar effect as stratigraphic uncertainties and can be modelled with the Monte-Carlo simulations applied in this study. As sedimentation rates always vary within a sedimentary series, any particular astronomical cycle can be recorded on various thicknesses of sediments, which in turn decreases the power of this astronomical cycle and distributes its power over a large range of frequencies (Weedon, 2003). Stratigraphic uncertainties thus add additional noise which blurs the spectra of sedimentary series at high frequencies. Astronomical tuning can help in removing the effects of stratigraphic uncertainties and variations in sedimentation rates (e.g. Hays et al., 1976; Huang et al., 2010; Zeeden et al., 2013). The identification of the repetition of any astronomical cycle and their attribution to the same duration removes the effects of distortion of the

sedimentary series, and concentrates the variance of the power over several frequencies. Filtering a band of frequencies of interest can help in identifying the repetition of the cycle used for the astronomical calibration (e.g. Westerhold et al., 2008; Thibault et al., 2016; De Vleeschouwer et al., 2015). Because of distortions of the sedimentary series, a filter, if designed very narrowly, can lead to a distortion of the actual amplitude and number of repetitions of the filtered frequency. This is particularly critical for the precession band, which has been proven to be sensitive to stratigraphic uncertainty (Figs. 8 to 11), and for which amplitude modulation is governed by eccentricity. The use of a wide-band filter, such as in the procedure of Zeeden et al. (2015), limits these biases and helps in a better reconstruction of the short wavelengths. Otherwise, a robust reconstruction of the amplitude modulation of the precession band requires limited biases of the power spectrum in the precession, which requires a good control on the sample position in the field. In addition, the simulations indicate that taking at least 4-10 samples per cycle should allow calculation of robust power spectra estimates in the respective cycle band (Table 1; Figs. 8-11).

Also in the evaluation of the relative contribution of precession and obliquity-related climatic forcing, an accurate assessment of the respective spectral power is essential (Ghirardi et al., 2014; Latta et al., 2006; Martinez et al., 2013; Weedon et al., 2004). Notably, whenever obliquity cycles are expressed more strongly compared to precession cycles, this has been interpreted as a reflection of important climate dynamics and feedback mechanisms at high latitudes (Ruddiman and McIntyre, 1984), the build-up and decay of quasi-stable carbon reservoirs (Laurin et al., 2015), or direct obliquity forcing at tropical latitudes (Bosmans et al., 2015; Park and Oglesby, 1991). A robust evaluation of the relative contribution of precession and obliquity requires at least that no bias occurs from the generation of the depth series, which includes the sampling procedure. This is particularly crucial in the case where the autoregressive coefficient of the red-noise background is high as in the La Thure series. Because of their low powers in the spectrum of the raw series, the spectral peaks related to the precession cycles become not significant at 10 to 15% uncertainties (Figs. 9-10). In that case, one can misleadingly interpret the absence of the record of the precession cycles in the sedimentary series while the absence of significant high frequencies can simply be the consequence of spectral smoothing when increasing the level of stratigraphic uncertainty. Once again, a good control of the sample position accompanied by a high density of sampling will importantly improve interpretations of the relative contributions of the precession and obliquity on the spectrum, which will in turn help making accurate palaeoclimatic interpretations.

## 8  Conclusions

Errors made during the measurement of the stratigraphic position of a sample significantly affect the power spectrum of depth series. We present a method to assess the impact of such errors that is compatible with different techniques for spectral analysis. Our method is based on a Monte-Carlo

procedure that randomises the sample steps of the time series, using a gamma distribution. Such a distribution preserves the stratigraphic order of samples, and allows controlling the average and

515 the variance of the distribution of sample steps after randomisation. The simulations presented in this paper show that the gamma distribution of sample steps realistically simulates errors that are generally made during the measurement of sample positions. The three case studies presented in this paper all show a strong decrease in the power spectrum at high frequencies. Simulations indicate that the power spectrum can be completely smoothed for periods less than 3-4 times the average sample

distance. Thus taking at least 3-4 samples per thinnest cycle of interest (e.g., precession cycles for the Milankovitch band) should preserve spectral peaks of this cycle. However, the decrease of power observed in a large portion of the spectrum implies a decrease in the significance level of the spectral peaks. Taking at least 4-10 samples per thinnest cycle of interest should allow their significance level to be preserved, depending on the level of stratigraphic uncertainty and depending on the redness of

the power spectrum. Robust reconstruction of the power spectrum in the entire Milankovitch band requires a robust control of the sample step in the field, and requires a high density of sampling. To avoid any dispersion of the power spectrum in the precession band, taking 10 samples per precession cycles appears to be a safe density of sampling. For lower resolution sampling, we recommend to apply gamma-law simulations to ensure that stratigraphic uncertainty only has limited impact on

the spectral power and significance level of the targeted cycles. Gamma-law simulations can also be used to simulate the effect of variations in the sedimentation rate on insolation series, which should help in modelling the transfer from insolation series to sedimentary series.

*Acknowledgements.* ERC Consolidator Grant "EarthSequencing" (Grant Agreement No. 617462) funded this project. We acknowledge Christian Zeeden and Linda Hinnov for their thorough and insightful reviews. We

also thank Anna-Joy Drury for English-proof reading.

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

**Appendix A: Optimized linear interpolation**

When interpolating an unevenly sampled time-series to an even sample distance, part of amplitude is
lost in the high frequencies because the sample positions in the interpolated series do not necessarily
correspond to the position of the maxima and minima of the original time-series (Figs. A1a and
b). Oversampling has been suggested to limit the loss of amplitude during the interpolation process
(Hinnov et al., 2002). However, oversampling impacts the autoregressive coefficient when estimating
the level of red noise in the spectrum background (Hinnov, 2016). The optimized linear interpolation
used here is designed to limit the loss of amplitude of high-frequency cycles by finding the best-fit
between the original and the resampled time series (Fig. A1c, Eq. A1):

$$M = \frac{1}{n} \cdot \sum_{i=1}^{n} |s_{ori}[i] - s_{interp}[i]| \tag{A1}$$

with $M$ is the average misfit between the variable values of the two curves, $n$ is the number of points
compared, $s_{ori}$ is the original signal, and $s_{interp}$ is the resampled signal at the average sample
distance of the original series. This comparison is only possible if the depths (or ages) of $s_{ori}[i]$ and
$s_{interp}[i]$ are the same. This is of course not the case between the original and the resampled time
series (Fig. A1b), otherwise interpolation would not be necessary. To circumvent this problem, the
original and the resampled time series are both linearly interpolated with a sample step equal to the
maximum resolution by which the depths (or ages) are provided. For instance, in the case of the La
Thure series, the depths are given with a resolution of 0.01 m, so that $s_{ori}$ and $s_{interp}$ are linearly
interpolated at 0.01 m. This procedure does not change the shape of neither the original time series
nor the time series resampled at the average sample distance (Fig. A1c).

To test which resampled time series fits best with the original time series, various depths are
tested as starting points to resample the time at the average sample distance (Fig. A1d). The various
scenarios of starting points tested increment by $dx$ and have the following range:

$$T_{st.test} = T_{st.ori} : dx : (T_{st.ori} + dmoy - dx) \tag{A2}$$

with $T_{st.test}$ the tested starting points of time series resampled at the average sample distance, $T_{st.ori}$
the starting point of the original time series, $dmoy$ the average sample distance of the original time
series, and $dx$ the resolution with which the depths (or ages) are given.
The best-fit curve is the one for which M is minimized.

An example of application is shown for the La Thure section in Fig. A2. Differences in the result-
ing spectrum between the best-fit and the worst-fit resampled time series are displayed in this figure.
Main differences in the spectra of the two cases are observed in the middle and high frequencies.
Compared to the worst-fit resampling, the spectra of the best-fit resampling show decreased power
and confidence levels in the middle frequencies (from 0.2 to 0.7 cycles m$^{-1}$), while increased power
and confidence levels rather occur in the high frequencies (from 0.7 cycles m$^{-1}$ to the Nyquist fre-

quency). Fitting the best curve to the original time series thus impacts on the calculation of the power spectrum and the confidence levels of the spectral peaks.

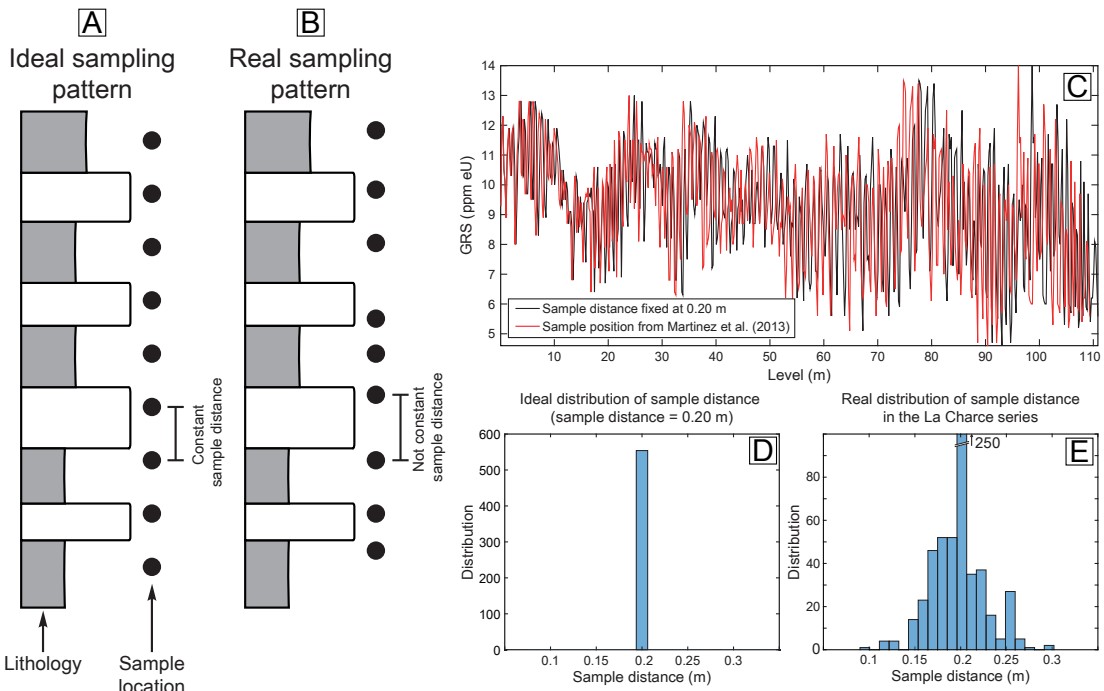

**Figure 1.** Illustration of the problem. (a) Theoretical sedimentary log with position of samples in an ideal case where the samples are strictly equally distant. (b) Theoretical sedimentary log with position of samples in a common sampling pattern where all samples are not strictly equally distant. Here the error in the sample position is exaggerated for the purpose of the example. (c) The gamma-ray series from La Charce shown as if all samples were strictly equidistant (black curve), and as they are positioned in Martinez et al. (2013) (red curve). (d) Distribution of sample distances in case of ideal sampling of the La Charce series (all sample distances are fixed at 0.20 m). (e) Distribution of sample distances in case of the La Charce series as published in Martinez et al. (2013).

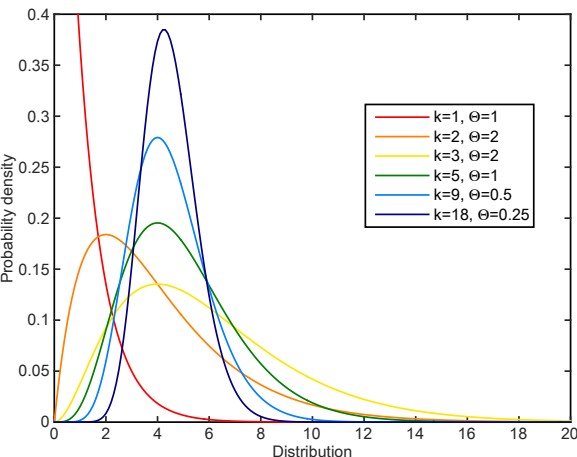

**Figure 2.** Gamma probability density functions (PDF). All Gamma PDF's have a positive support, which is a crucial characteristic to realistically simulate sample steps. The gamma density probability functions were generated with the Matlab *gampdf* function.

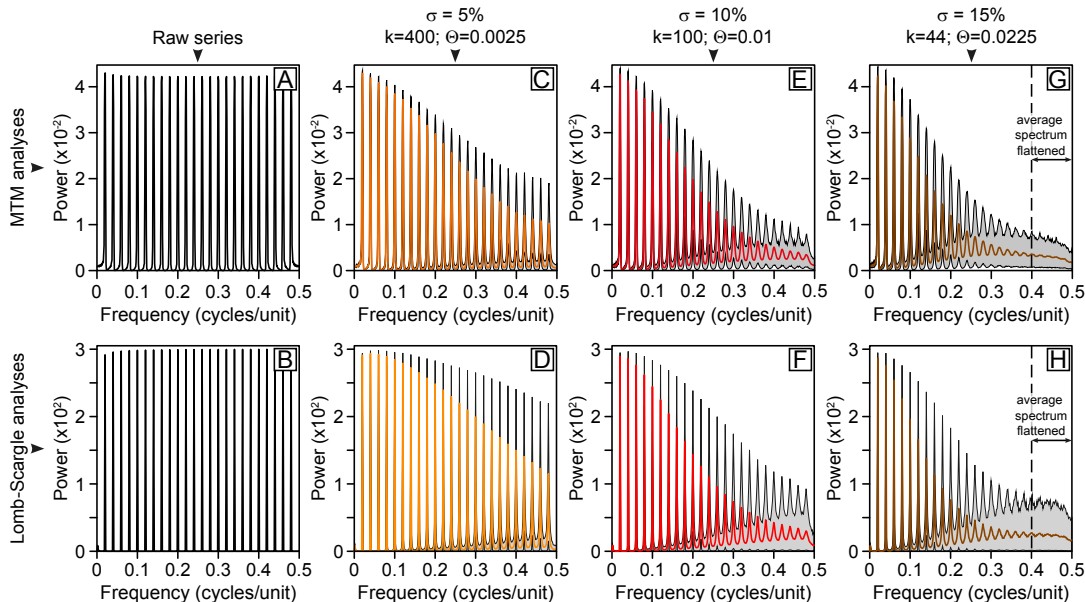

**Figure 3.** Effect of the gamma-law randomised sample distances on the $2\pi$-MTM and Lomb-Scargle spectra of the series of sum of pure sinusoids. (a) and (b) Spectra of the series without sample step randomisation. (c) and (d) with 5% of stratigraphic uncertainty. (e) and (f) with 10% of stratigraphic uncertainty. (g) and (h) with 15% of stratigraphic uncertainty. For each simulation shown from (c) to (h), the grey area represents the interval covering 95% of the simulations, while the red, orange and brown curves represent the average spectrum.

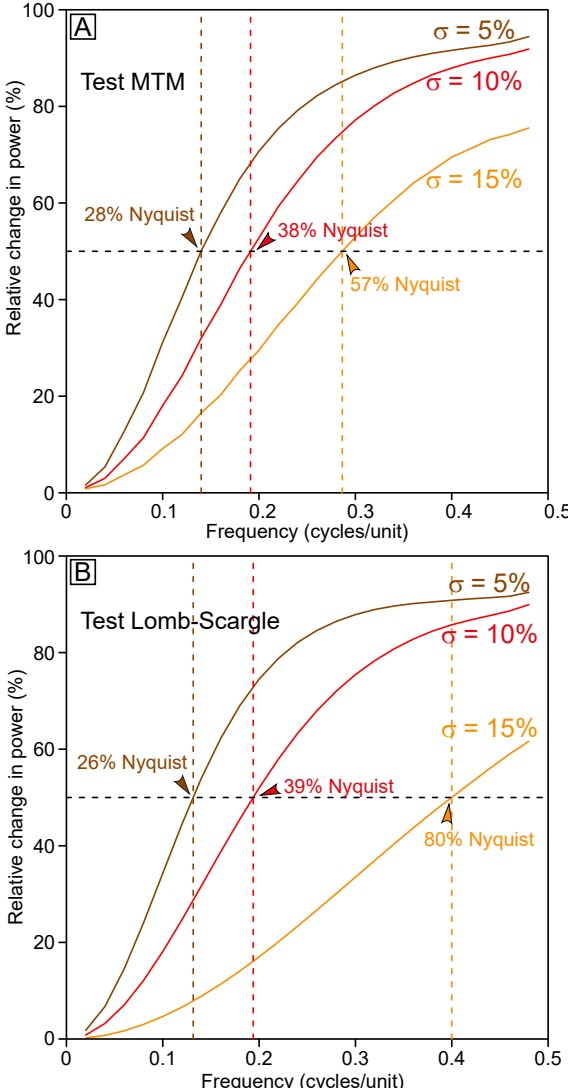

**Figure 4.** Relative change in power in the (a) $2\pi$-MTM spectra, and (b) Lomb-Scargle spectra after applying the gamma-law simulations of distortion of the time series. The arrows indicate at which frequency (relatively to the Nyquist frequency) the change in power reaches 50%.

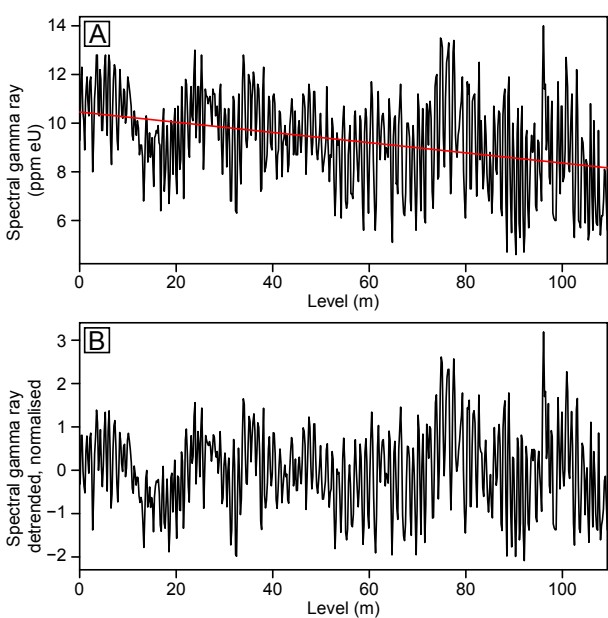

**Figure 5.** Detrending procedure of the gamma-ray series from the La Charce section. (a) Raw gamma-ray signal (black curve) with best-fit linear trend (red curve). (b) Gamma-ray series after subtraction of the linear trend and standardisation (average = 0; standard deviation = 1).

**Table 1.** Results of red-noise background estimates from the La Charce and the La Thure series with the $2\pi$-MTM and the REDFIT analyses.

|  |  | $\sigma = 0\%$ | $\sigma = 5\%$ | $\sigma = 10\%$ | $\sigma = 15\%$ |
|---|---|---|---|---|---|
| **La Charce MTM** | Autoregressive coefficient | 0.440 | $0.433 \pm 0.025$ | $0.432 \pm 0.037$ | $0.434 \pm 0.048$ |
|  | Average power ($\cdot 10^{-4}$) | 3.54 | $3.55 \pm 0.13$ | $3.58 \pm 0.20$ | $3.61 \pm 0.25$ |
| **La Charce REDFIT** | Autoregressive coefficient | 0.468 | $0.468 \pm 0.002$ | $0.467 \pm 0.003$ | $0.467 \pm 0.006$ |
|  | Average power | 0.398 | $0.399 \pm 0.003$ | $0.402 \pm 0.005$ | $0.407 \pm 0.008$ |
| **La Thure MTM** | Autoregressive coefficient | 0.657 | $0.658 \pm 0.025$ | $0.653 \pm 0.029$ | $0.651 \pm 0.033$ |
|  | Average power ($\cdot 10^{-3}$) | 1.67 | $1.67 \pm 0.04$ | $1.67 \pm 0.05$ | $1.68 \pm 0.07$ |
| **La Thure REDFIT** | Autoregressive coefficient | 0.407 | $0.406 \pm 0.004$ | $0.405 \pm 0.008$ | $0.404 \pm 0.013$ |
|  | Average power | 0.890 | $0.894 \pm 0.011$ | $0.900 \pm 0.019$ | $0.904 \pm 0.027$ |

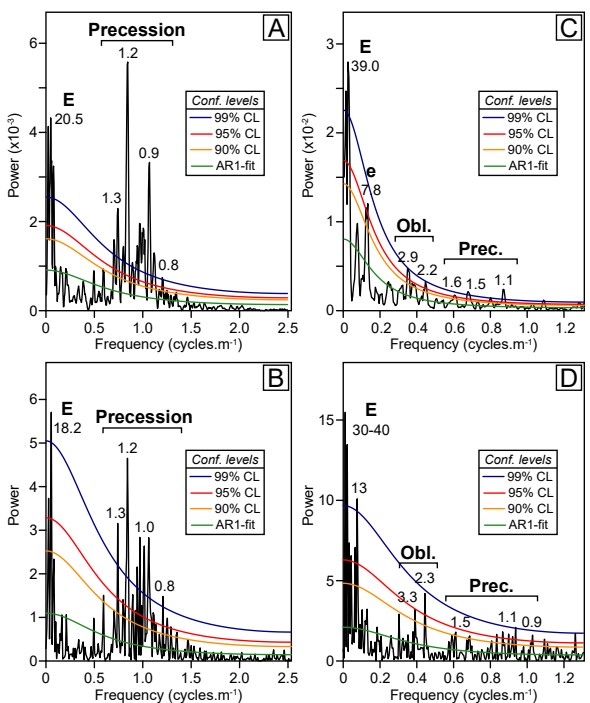

**Figure 6.** Spectra of the La Charce and La Thure series before Monte-Carlo simulations of the sample distances. (a) $2\pi$-MTM spectrum of the La Charce series. (b) REDFIT spectrum of the La Charce series. (c) $2\pi$-MTM spectrum of the La Thure series. (d) REDFIT spectrum of the La Thure series. The main significant periods are given in meters.

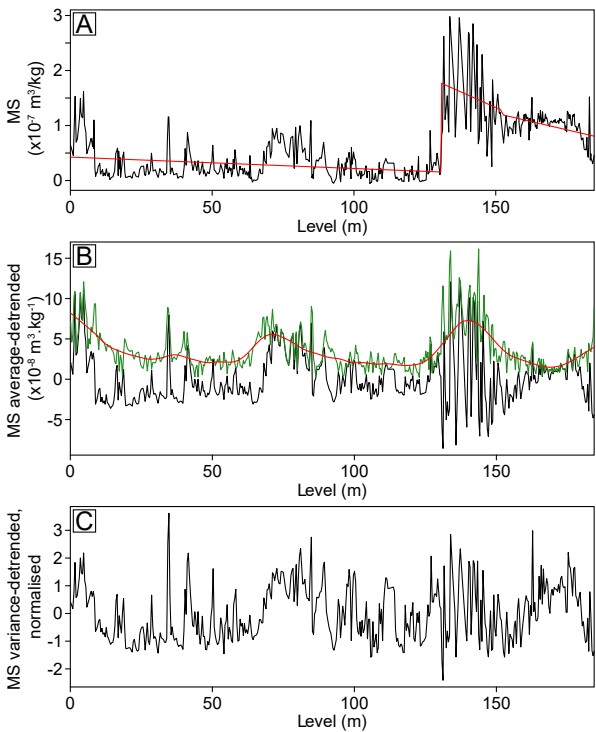

**Figure 7.** Detrending procedure of the magnetic susceptibility (MS) series from the La Thure section. (a) Raw MS signal (black curve) with piecewise best-fit linear trend of the average (red curve). (b) MS series after subtraction of the piecewise linear trend (black curve), with instantaneous amplitude (green curve) and LOWESS regression of the instantaneous amplitude applied with a coefficient of 10% (red curve). (c) MS curve after dividing the MS series "average-detrended" by the LOWESS regression of the instantaneous amplitude, and after standardisation.

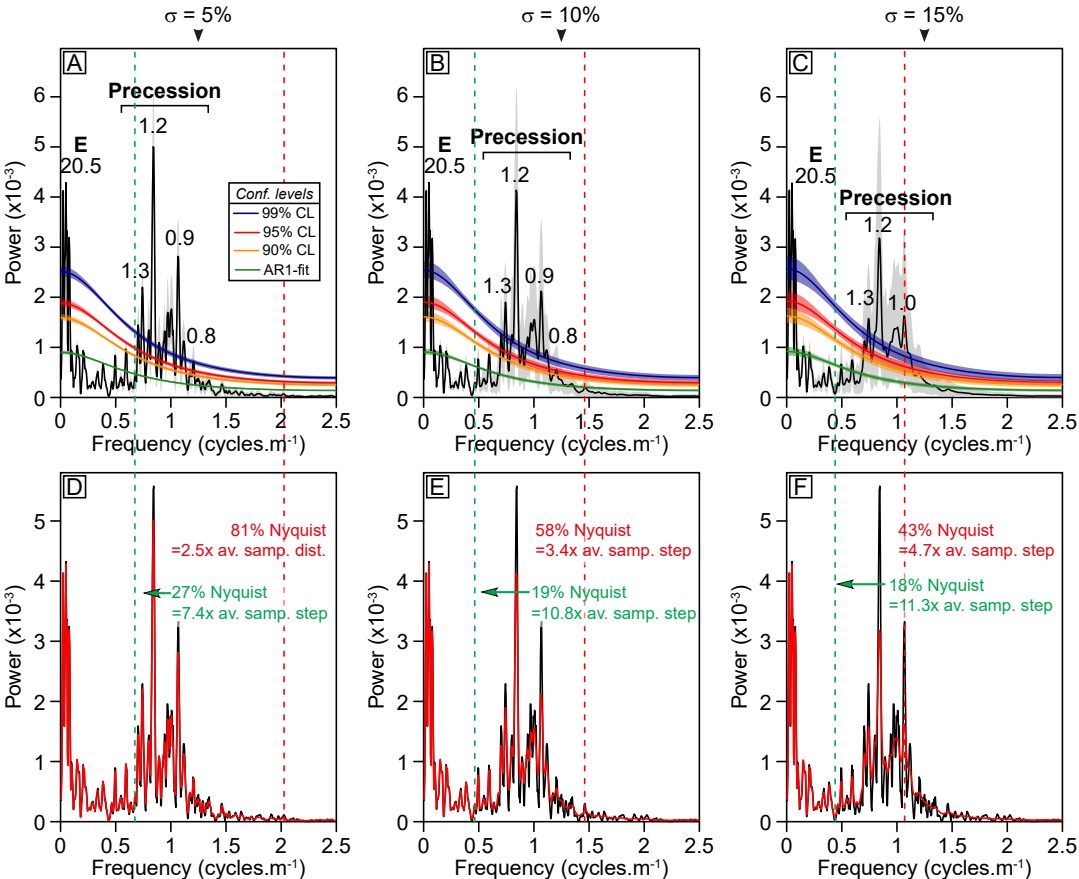

**Figure 8.** Effect of the gamma-law randomisation of the sample distances on the $2\pi$-MTM spectrum of the La Charce series. (a to c) $2\pi$-MTM spectra with a level of stratigraphic uncertainty fixed to 5%, 10% and 15% of the average sample distance of the series. The grey area represents the interval covering 95% of the simulations. The average confidence levels are reported on the spectra with their respective areas covering 95% of the simulations. Main significant periods are indicated in meters with, in bold, their corresponding orbital cycles. E: 405-kyr eccentricity. (d to f) Superposition of the $2\pi$-MTM spectra before randomisation (in black) and the average spectrum after the 1000 simulations (in red). The red dashed lines indicate the lowest frequency above which the spectrum is completely smoothed, so that no more frequency can be identified. The green dashed lines represent the highest frequency below which the spectrum of the series before randomisation appears practically identical to the spectrum after randomisation.

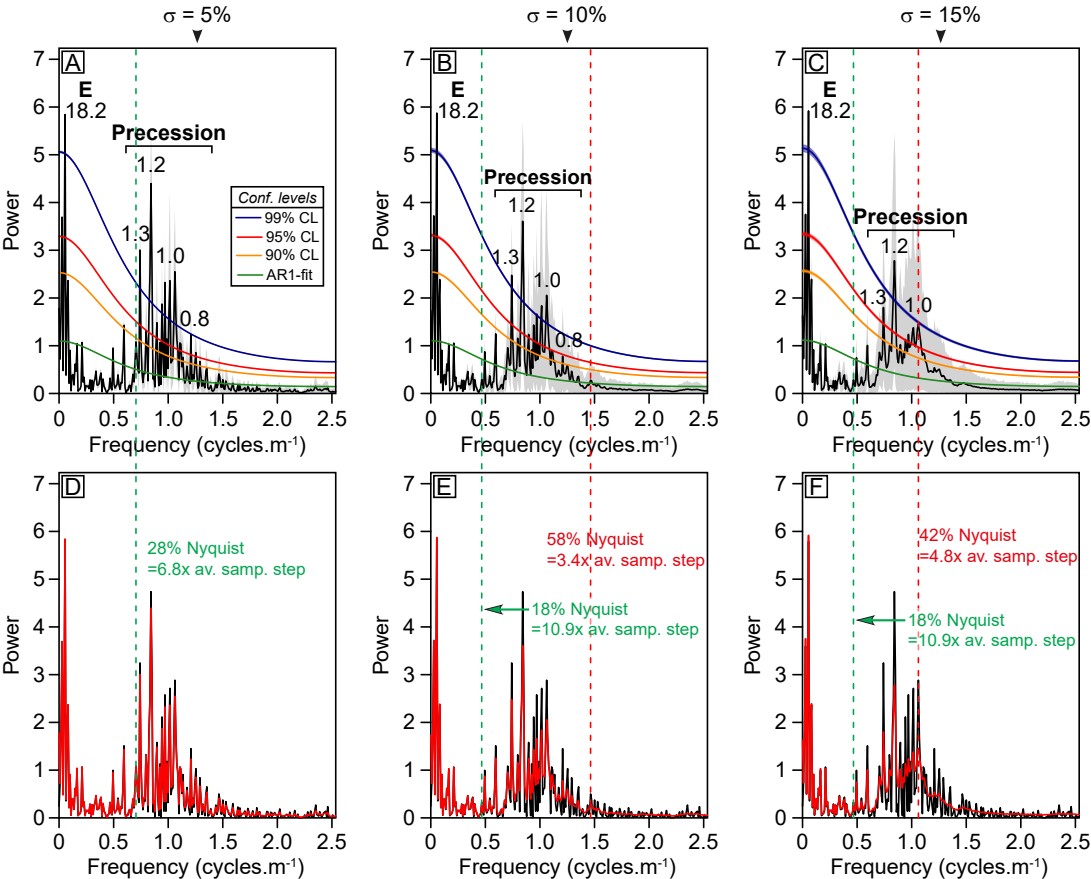

**Figure 9.** Effect of the gamma-law randomisation of the sample distances on the REDFIT spectrum of the La Charce series. (a to c) REDFIT spectra with a level of stratigraphic uncertainty fixed to 5%, 10% and 15% of the average sample distance of the series. The grey area represents the interval covering 95% of the simulations. The average confidence levels are reported on the spectra with their respective areas covering 95% of the simulations. Main significant periods are indicated in meters with, in bold, their corresponding orbital cycles. E: 405-kyr eccentricity. (d to f) Superposition of the REDFIT spectra before randomisation (in black) and the average spectrum after the 1000 simulations (in red). The red dashed lines indicate the lowest frequency above which the spectrum is completely smoothed, so that no more frequency can be identified. The green dashed lines represent the highest frequency below which the spectrum of the series before randomisation appears practically identical to the spectrum after randomisation.

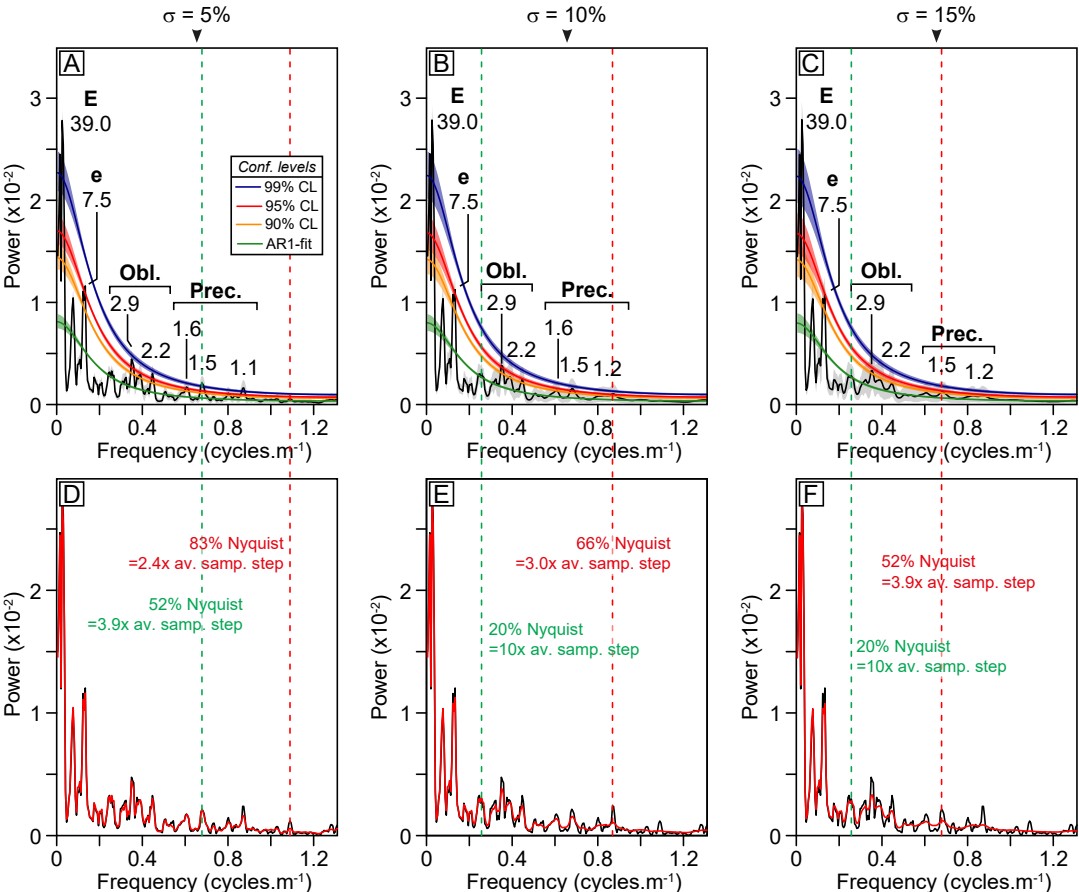

**Figure 10.** Effect of the gamma-law randomisation of the sample distances on the $2\pi$-MTM spectrum of the La Thure series. (a to c) $2\pi$-MTM spectra with a level of stratigraphic uncertainty fixed to 5%, 10% and 15% of the average sample distance of the series. The grey area represents the interval covering 95% of the simulations. The average confidence levels are reported on the spectra with their respective areas covering 95% of the simulations. Main significant periods are indicated in meters with, in bold, their corresponding orbital cycles. E: 405-kyr eccentricity; e: 100-kyr eccentricity. (d to f) Superposition of the $2\pi$-MTM spectra before randomisation (in black) and the average spectrum after the 1000 simulations (in red). The red dashed lines indicate the lowest frequency above which the spectrum is completely smoothed, so that no more frequency can be identified. The green dashed lines represent the highest frequency below which the spectrum of the series before randomisation appears practically identical to the spectrum after randomisation.

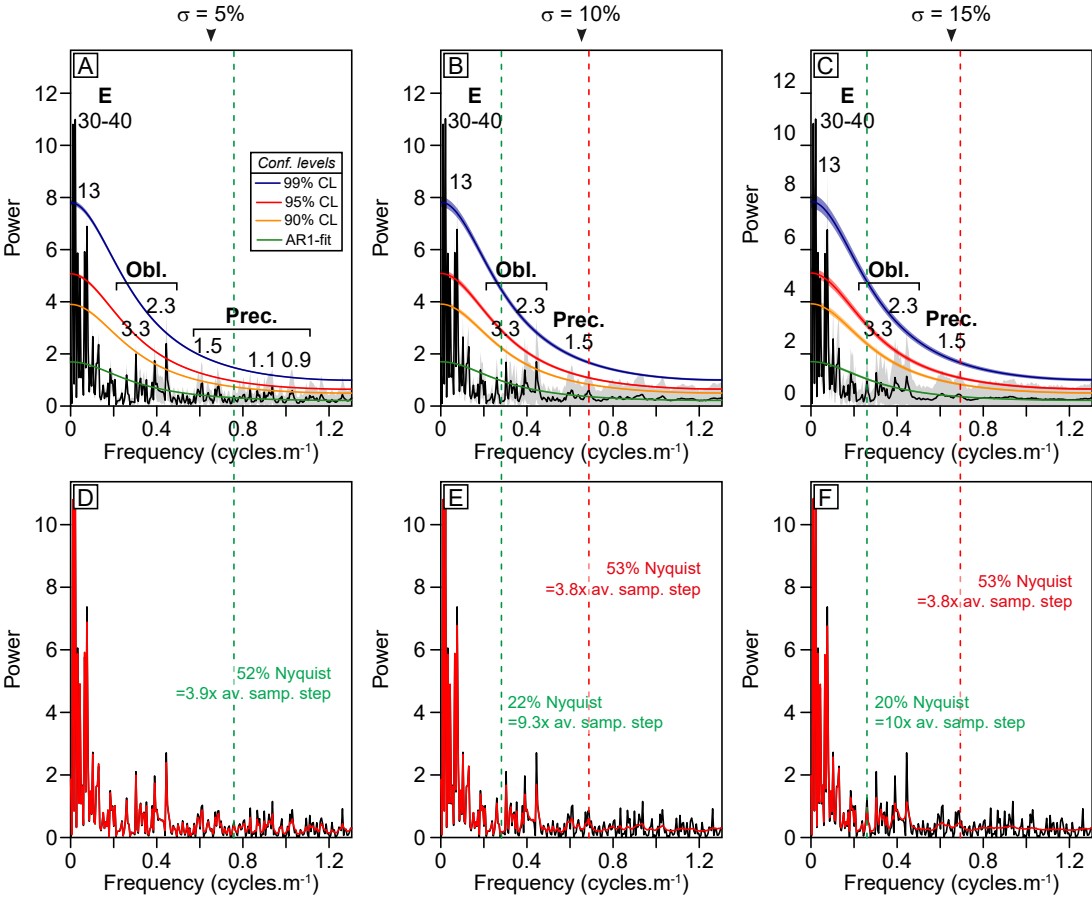

**Figure 11.** Effect of the gamma-law randomisation of the sample distances on the REDFIT spectrum of the La Charce series. (a to c) REDFIT spectra with a level of stratigraphic uncertainty fixed to 5%, 10% and 15% of the average sample distance of the series. The grey area represents the interval covering 95% of the simulations. The average confidence levels are reported on the spectra with their respective areas covering 95% of the simulations. Main significant periods are indicated in meters with, in bold, their corresponding orbital cycles. E: 405-kyr eccentricity; e: 100-kyr eccentricity. (d to f) Superposition of the REDFIT spectra before randomisation (in black) and the average spectrum after the 1000 simulations (in red). The red dashed lines indicate the lowest frequency above which the spectrum is completely smoothed, so that no more frequency can be identified. The green dashed lines represent the highest frequency below which the spectrum of the series before randomisation appears practically identical to the spectrum after randomisation.

**Table 2.** Synthesis of the results of highest frequencies before smoothing of the spectra when applying the Monte-Carlo simulations, and of highest frequency in which the spectra before and after simulation are practically identical.

| | | Level of stratigraphic uncertainty | | |
| --- | --- | --- | --- | --- |
| | | $\sigma = 5\%$ | $\sigma = 10\%$ | $\sigma = 15\%$ |
| **La Charce MTM** | Highest frequency before smoothing | 81% Nyquist | 58% Nyquist | 43% Nyquist |
| | Equivalent number sample steps | 2.5x | 3.4x | 4.7x |
| | Highest frequency confounded spectra | 27% Nyquist | 19% Nyquist | 18% Nyquist |
| | Equivalent number sample steps | 7.4x | 10.8x | 11.3x |
| **La Charce REDFIT** | Highest frequency before smoothing | | 58% Nyquist | 42% Nyquist |
| | Equivalent number sample steps | | 3.4x | 4.8x |
| | Highest frequency confounded spectra | 28% Nyquist | 18% Nyquist | 18% Nyquist |
| | Equivalent number sample steps | 6.8x | 10.9x | 10.9x |
| **La Thure MTM** | Highest frequency before smoothing | 83% Nyquist | 66% Nyquist | 52% Nyquist |
| | Equivalent number sample steps | 2.4x | 3.0x | 3.9x |
| | Highest frequency confounded spectra | 52% Nyquist | 20% Nyquist | 20% Nyquist |
| | Equivalent number sample steps | 3.9x | 10x | 10x |
| **La Thure REDFIT** | Highest frequency before smoothing | | 53% Nyquist | 53% Nyquist |
| | Equivalent number sample steps | | 3.8x | 3.8x |
| | Highest frequency confounded spectra | 52% Nyquist | 22% Nyquist | 20% Nyquist |
| | Equivalent number sample steps | 3.9x | 9.3x | 10x |

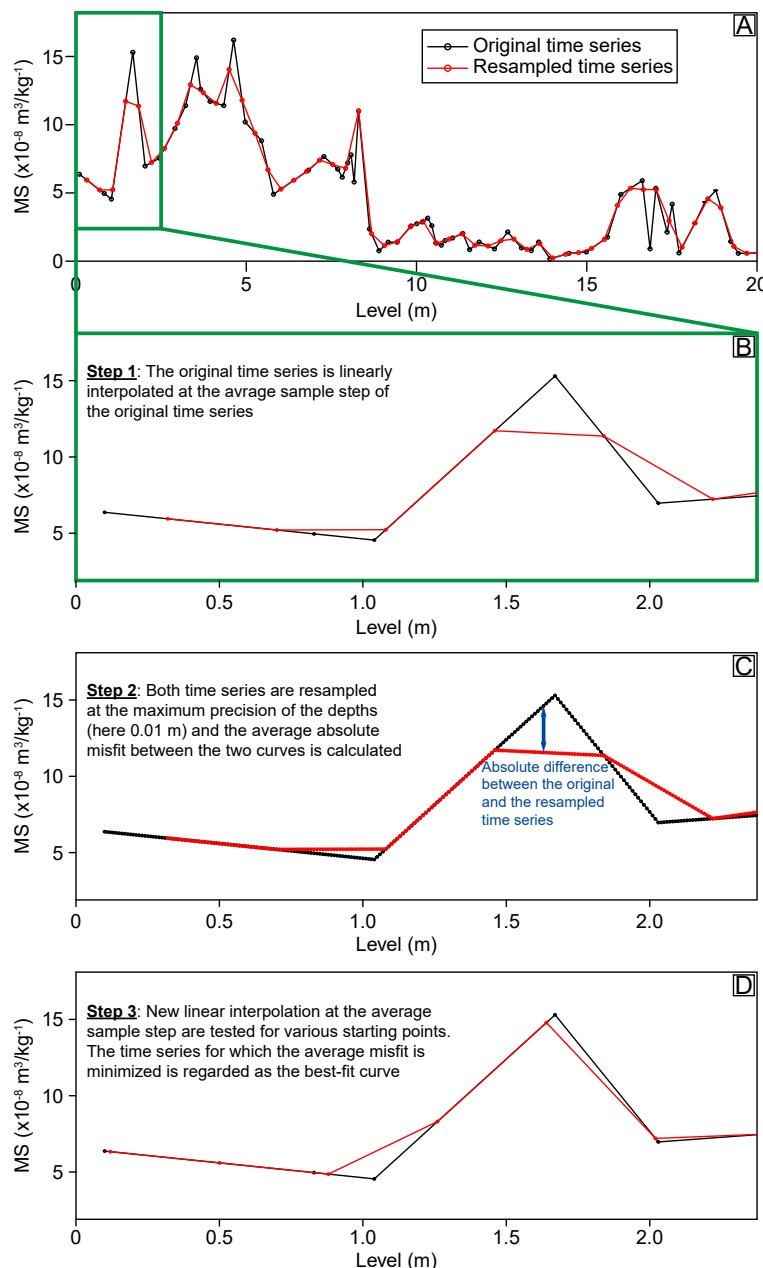

**Figure A.1.** Scheme of the procedure of the optimized linear interpolation of time series.

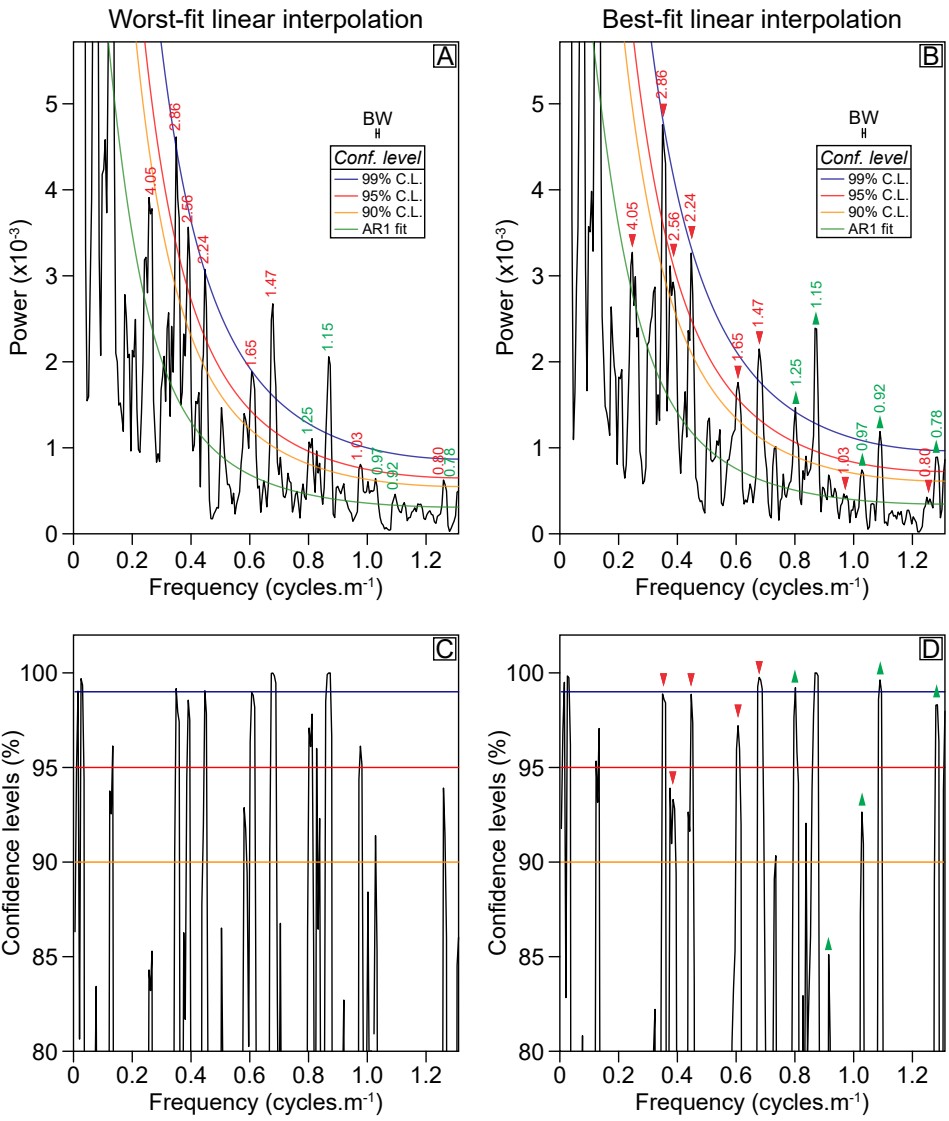

**Figure A.2.** Comparison of spectra of the resampled time series for the worst-fit case (a and c), and for the best-fit case (b and d). Spectra (a) and (b) are calculated using the $2\pi$-multitaper method with confidence levels calculated using the method of Mann and Lees (1996) with a Tukey's end-point rule (Meyers, 2014). (c) and (d) show the confidence levels compared to a red noise. Red arrows indicate the frequency at which powers and confidence levels decrease from the worst-fit case to the best-fit case. Green arrows display the opposite case.