# Peer review of "Testing the impact of stratigraphic uncertainty on spectral analyses of sedimentary series"

_Climate of the Past, 2015_

## Referee Comment (RC1) · C. Zeeden (Referee) · 25 Feb 2016

Dear Dr. Martinez, Dear authors,

You submitted a manuscript entitled 'Testing the impact of stratigraphic uncertainty on spectral analyses of sedimentary series'. I was asked to review your paper, and especially pay attention whether it is suitable for climate of the past because it is quite technical.

In your manuscript you focus on a very fundamental part of spectral analysis – the sampling, and the effect of non-equally spaced sampling for spectral analysis. Using simulations you demonstrate the effect of not precisely equally spaced sampling, and make applicable suggestions for sampling strategies. Your methods are solving a scientific question and are chosen logically. These are novel aspects and without doubt

publishing prominently in the stratigraphy and paleoclimatology community. In my opinion your paper is clearly written and well structured. Figures supplement the text in a logical way.

Your paper is indeed technical, but at the same time of fundamental importance for the interpretation of (semi)cyclic paleoclimate signals. You outline all necessary technical details, and take also non-expert readers through your manuscript. In my opinion the manuscript is suitable for publication in Climate of the Past. However, making several rather minor adjustments can make the manuscript more accessible to a wider audience, specifically I would suggest:

- focus on more applicable and less technical results in the abstract and conclusions. The technical details are important, and you outline them well. However, I would suggest to be less technical specifically in the abstract and conclusions. Highlight that effect of sampling uncertainty alters power spectra, and that generally precession will be more affected than obliquity and eccentricity. I would suggest to directly stating that sampling uncertainty can have an effect on interpretations derived from relative precession- and obliquity power.

- the La Thure series shows both precession and obliquity. Could you exemplary discuss what the result from your test means for this example record, and how it aids the interpretation?

- explain what the Nyquist frequency represents.

Further I would suggest you to clarify several points:

- you suggest uncertainty to be fully random. I propose to briefly discuss why you assume this – and what effect(s) systematic uncertainty may have.

Lines 117-120: 106-116m is the overall spread in section thickness. From a conceptual point of view I think that this spread can hardly directly be used to estimate uncertainty in sample distance, because you see a result of $\sim$550 (gamma distributed) sample

distances summed up. Several of these will be shorter and longer than 20 cm – so your relative uncertainty will probably be higher – or fully systematic.

Lines 143, 223: Do I understand correct that you interpolate all time series (also with spacing of ∼0.2 m and ∼0.38 m) at 0.01 m intervals? Is this necessary and useful, and does this oversampling influence your results?

Lines 159-164: Your approach is good, but personally I would propose to also determine 95% confidence intervals of power by considering not only the average power spectrum from simulations. This may facilitate to compare (integrated) precession and obliquity power for paleoclimate studies.

175ff: a table summarizing the results presented may be helpful in addition. - In Fig. 4 the confidence levels of the MTM and Lomb-Scargle spectra are different. I would propose to mention this in the figure caption.

Line by line comments which may impriove the manuscript:

10, 13: maybe express Nyquist frequency as sampling interval to be clearer

15-17: "In addition, the simulations indicate that taking at least 6-10 samples per precession cycle should allow calculation of robust power spectra estimates in the Milankovitch band." – This is not limited to precession I think, what about a more general statement as 'In addition, the simulations indicate that taking at least 6-10 samples per cycle should allow calculation of robust power spectra estimates in the respective cycle band'?

28-29: "In core sediments, uncertainties in the sample position are also observed when performing physical sampling at very high resolution or because of core expansion phenomena (Hagelberg et al., 1995)" – suggestion: 'In cored sediments, uncertainties in the sample position are also observed when performing physical sampling at very high resolution or because of core expansion phenomena (Hagelberg et al., 1995) or imperfect coring (Ruddiman et al., 1987).'

37-38: "In this study, we address this problem by quantifying the impact of such errors on the frequency, as well as the power of higher-frequency cycles." → the second part of this sentence ("the frequency, as well as the power of higher-frequency cycles") may be 'the frequency and power distributions'?

42-44: This sentence seems in contradiction to the last sentence of the abstract, more consistent phrasing may solve this.

48: delete 'correctly'?

64: remove 'easily'

98/99: could you mention that these are Devonian, and give a rough age as for the La Charce section?

108: are the two brackets necessary?

119/120: "with an average of 110.3 ± 5.1 m, and a relative uncertainty of 4.6%" I would propose to mention that the "5.1 m" and "uncertainty of 4.6%" are estimated from only three experiments, and that these are regarded as representative, but may not be actually.

146: maybe give also reference to the R package used ('dplR')

155: "The confidence levels of the datasets were calculated before randomisation and directly plotted to the simulated spectra." I am unsure how this is meant, and I would suggest phrasing this more clearly.

160-164: "Pori: the power spectrum before randomization" – as you calculate this for individual frequencies, following may be more clear: 'Pori: power before randomization for a specific frequency', same for Pave (if I understand this correct).

172 "with 5% uncertainty" – maybe clarify as 'with 5% stratigraphic uncertainty'

200: with "first frequency" 'lowest frequency' is meant I assume – could you clarify this?

205-211: Please make clearer that geological data usually have no precise frequencies, but frequency ranges. You mention this, but I am not sure if everyone will understand this easily.

217: I am not sure if you need to mention "that the stratigraphic order of the samples in the raw series is preserved after randomisation" again. You develop this earlier in the manuscript.

220f: "This difference realistically simulates small thickness errors, which accumulate when measuring successive sample steps." – this can in my opinion be formulated better, and should highlight that errors may accumulate, or may also not accumulate but level out.

241: "above 40% of the Nyquist frequency", I would suggest to also mention the frequency, maybe in brackets after this statement. Maybe bring these ratios in direct reference to precession (e.g. ~1/3rd of precession frequency/wavelength), so that this is more clear for readers not so familiar with time series analysis.

258/59: "As in the case of the La Charce series, the stratigraphic order of the samples is preserved in the randomised series" – In my opinion this is clear by now in the manuscript, and does not need to be repeated.

304: replace "powers" by "power"

310: "result suggest" – one of these need an "s" in the end

312/13: "This requires that more than 6 samples per precession cycle have to be taken" - samples or measurements?

355: 356: maybe also refer to (Meyers, 2015; Shackleton et al., 1995)

396: "on the field" – in the field?

Additional References Cited

Meyers, S. R. : The evaluation of eccentricity-related amplitude modulation and bundling in paleoclimate data: An inverse approach for astrochronologic testing and time scale optimization, Paleoceanography, 2015, PA002850, doi:10.1002/2015PA002850, 2015.

Ruddiman, W. F., Cameron, D. and Clement, B. M.: Sediment Disturbance and Correlation of Offset Holes Drilled with the Hydraulic Piston Corer - Leg 94, Initial Reports of the Deep Sea Drilling Project, 94, 615–634, 1987.

Shackleton, N. J., Hagelberg, T. K. and Crowhurst, S. J.: Evaluating the success of astronomical tuning: Pitfalls of using coherence as a criterion for assessing pre-Pleistocene timescales, Paleoceanography, 10(4), 693–697, doi:10.1029/95PA01454, 1995.

---

## Referee Comment (RC2) · L. A. Hinnov (Referee) · 24 Mar 2016

**1. Introduction**

Martinez et al. explore the problem of uncertainties that arise from the intersection of variable sedimentation rate and sampling errors in the analysis of Milankovitch-forced stratigraphy. This work provides important guidance for stratigraphers faced with decisions on how to sample cyclic sedimentary sequences in a way that optimizes recovery of paleoclimate signals. This represents a significant contribution to the study of paleoclimate spectra.

**2. The error model**

The authors call on the gamma probability density distribution to characterize strati-

graphic sampling. Here there could be more explanation, e.g., a simple illustration of the problem, i.e., in Figure 1 add a diagram of a hypothetical stratigraphic section, different sampling sequences, and their histograms – perhaps the same ones as presented in Figure 2); in Figure 1 caption indicate "gampdf(x, k, $\Theta$)" and label horizontal axis as "x". The models presented in Figure 2 displayed in F, G and H: what values of k and $\Theta$ do these correspond to?

3. The geological datasets

Two typical cases are presented, from Cretaceous and Devonian cyclostratigraphic outcrops that were previously sampled and analyzed, with publicly available datasets. This allows the reader to directly replicate the uncertainty modeling presented in this paper, for use as a template for other datasets and parameters.

4. Implementation of the models in the stratigraphic-uncertainty tests

This reviewer can personally attest to the difficulty in measuring a consistent thickness for the same outcrop by different researchers - in my experience in one case: 112 m vs. 132 m! For overturned sections, any dip error committed will contribute to a positive bias in stratigraphic thickness measurements. There is undoubtedly such a problem in the steeply dipping Cretaceous section at La Charce examined in this paper.

On issues concerning methods, it is important to restrict interpolation to mean or median rate when applying AR noise models with MTM spectra (such as used in SSA-MTM Toolkit). The Devonian section has a mean sample rate of 0.38 m – not clear what the median rate is – and this is much larger than the interpolation to 0.01 m. The Cretaceous section has a mean sample rate of 0.20 m, so has a similar problem. The authors should recalculate the MTM analysis with interpolation to the median sample spacing of the two sections. (The red noise spectra will be significantly different because of the way the autocorrelation lag-1 coefficient is calculated.) The other parameter that requires reporting is whether "log" or "linear" fitting was enabled in the calculation of robust red noise for the MTM spectra.

**5. Application to a sum of sinusoids**

This section quantifies the loss of power at high frequencies with increasing uncertainty of (variability in) the sample step sequence for a simulated sum-of-sinusoids series.

The absence of windowing in the Lomb-Scargle (LS) spectra would be expected to result in higher spectral variance compared to multitaper-windowed MTM spectra, and may account for the elevated grey spectra from the LS Monte Carlo simulations (compared to those of the MTM spectra). Interestingly, for 10% and 15% $\sigma$, loss of power occurs at practically the same frequencies in both MTM and LS spectra. Would it be possible to indicate the expected variance in Nyquist frequency for the 3 cases (5%, 10%, 15%) in order to understand the accuracy of the MTM and LS spectra?

A new order of the graphs in Figures 2 and 3 might benefit the presentation:

• New Figure 2: display Figs. 2F, G, H only, and explain how these relate to k and Θ (or put them into a Figure 1B). • New Figure 3: in top row, display Fig. 2A, B, C, D; bottom row display Fig. 3A, B, C, D. • Figs. 2E and 3E could be placed into a new figure.

What did we learn from this exercise and how will it help with the interpretation of the geological datasets to follow?

**6. Application to geological datasets**

The MTM spectrum of the Devonian series (Figure 4D) shows a robust red noise model with extremely elevated low frequencies, implying that a "log" fit was calculated in SSA-MTM Toolkit, and that the model suffers additionally from the 0.01 m interpolation (see comments for Section 4). Some of the text in this section about differences in red noise calculations (which by the way are not meaningfully explained) may not be needed once the interpolation problem is addressed.

**7. Discussion**

The main point of this study is that sampling is the critical decision that must be made when evaluating a stratigraphic sequence for paleoclimate signals. Almost all problems can be controlled with high-density sampling, e.g., 6-10 samples per putative precession cycle. It appears that one can easily expect 5% errors in stratigraphic position measurements, which combined with sedimentation rate variations, will mix the highest frequencies of a sampled sequence. Thus we are always alarmed at how low in power – and misaligned – precession cycles are in stratigraphic spectra. In the end, one never knows if a sample that has been collected has been assigned to its true stratigraphic position. This is an important limitation that is under-appreciated by the geological community and the authors should be commended for tackling this problem.

A number of issues have been left unexplored: (1) how does systematic sample position error, such as can occur with receded marls alternating with prominent limestones in outcrops, affect stratigraphic spectra; (2) can astronomical tuning bypass the positional uncertainty problem (notwithstanding the recent approach described in Zeeden et al., 2015); and (3) how does the positional uncertainty problem affect the red noise model estimates?

Other comments

Lines 23-24: The Multi-Taper Method (Thomson, 1982) might be more accurately characterized as a spectrum estimator that is based on the Fourier Transform – not as a derivative of the Fourier Transform.

Line 27: A recent massive improvement to the Jacob's staff in outcrop studies is terrestrial laser scanning with precision positioning at the mm level (Franceschi et al., 2011; Franceschi et al., 2015).

Line 101: Change to "Pas et al., 2015".

Lines 264- 265: what does the output of the "long-term trend of the variance" look like, and what was used to compute the "LOWESS regression with a 10% coefficient"?

[Figure]

Line 350: For monotonous stratigraphy yielding Milankovitch signal see also Latta et al., 2006.

Line 355: Change "require" to "requires"

Line 361: Delete "Note than".

Supplementary Fille: R package dplR appears to be used but not referenced in the main text. R is used to calculate REDFIT– is it provided in the dplR package?

References

Franceschi, M., Penasa, L., Coccioni, R., Gattacceca, J., Smit, J., Cascella, A., Mariani, S., and Montanari, A. (2015), Terrestrial Laser Scanner imaging for the cyclostratigraphy and astronomical tuning of the Ypresian–Lutetian pelagic section of Smirra (Umbria–Marche Basin, Italy), Palaeogeography, Palaeoclimatology, Palaeoecology, 440, 33–46, doi: 10.1016/j.palaeo.2015.08.027

Franceschi, M., Preto, N., Hinnov, L.A., Huang, C., and Rusciadelli, G. (2011), Terrestrial laser scanner imaging reveals astronomical forcing of the early Cretaceous Tethys realm, Earth and Planetary Science Letters, 305, 359-370, doi:10.1016/j.epsl.2011.03.017

Latta, D.K., Anastasio, D., Hinnov, L.A., Elrick, M.E., and Kodama, K.P. (2006), A record of Milankovitch rhythms in lithologically non-cyclic marine carbonates, Geology, 34, 29-32, doi: 10.1130/G21918.1

---

## Author Comment (AC1) · 22 Apr 2016

**Christian Zeeden (CZ): the La Thure series shows both precession and obliquity. Could you exemplary discuss what the result from your test means for this example record, and how it aids the interpretation?**
» The authors: This example is indeed interesting because both obliquity and precession have been observed (De Vleeschouwer et al., 2015), with obliquity having higher powers than precession. Several studies suggested that a dominance of obliquity in tropical sediments reflect cooling or icehouse conditions, while dominance of precession would be associated to greenhouse conditions (Zachos et al., 2001; Westerhold and Röhl, 2009; Boulila et al., 2011; ). With the implementation of our test we show that precession nearly vanishes at 15% of uncertainty. As a result, if one does not take into account this sampling bias on power spectrum one can misleadingly

interpret a dominance of obliquity in sediments, which impacts in turn on the climatic interpretations.

**CZ: explain what the Nyquist frequency represents.**
» The authors: The Nyquist frequency is the highest frequency (or smallest period) that can be detected. It corresponds to the inverse of twice the sample step. This information will be added in the next version of the manuscript.

**CZ: you suggest uncertainty to be fully random. I propose to briefly discuss why you assume this – and what effect(s) systematic uncertainty may have.**
» The authors: We assume a fully random error, based on comparisons with actual data of sample distances repeatedly measured on the La Charce series. This comparison went as follows: In a first step, the thicknesses of the individual beds were measured and a lithologic log was drawn based on these measurements. In a second step, samples were taken from the studied section every 20 centimetres and the sample positions were indicated on the lithologic log. After this second step, we observed that the distances between two successive samples was not exactly 20 cm, but rather ranged from 10 cm to 30 cm, with an average of 19.7 cm and a standard deviation of 2.5 cm. This observation was made by comparing the expected stratigraphic position of the nth sample (n x 20 cm) with the stratigraphic position of the bed the sample comes from in the lithologic log. The mismatch in sample position between the lithologic log and the bed from which the sample was taken can be quantified for every sample within the studied stratigraphic interval. We evaluated the distribution of every sample's mismatch and observed a log-normal distribution. This observation is the basis for our suggestion to consider the stratigraphic uncertainty to be fully random.
In addition, the total thickness of the series was measured at 109,33 m. With this thickness we expected to take 547 samples. Instead, we took 555 samples. We thus

have an error of 8 samples out of 555 samples, either 1.4% difference. This is of course much lower than the actual thickness measurement error for individual sample distances, which implies no systematic error.
We will briefly discuss the absence of systematic error in the revised manuscript

**CZ: Lines 117-120: 106-116m is the overall spread in section thickness. From a conceptual point of view I think that this spread can hardly directly be used to estimate uncertainty in sample distance, because you see a result of ∼550 (gamma distributed) sample distances summed up. Several of these will be shorter and longer than 20 cm – so your relative uncertainty will probably be higher – or fully systematic.**
» The authors: The reviewer is right. On average, the error made to measure the entire section is lower than the error made to measure sample distances. On entire sections, systematic errors will have for consequence overestimate the thickness of certain parts of the section while other parts will be have underestimated thicknesses. Thus, the error made to measure the total thickness of the section will be lower than the distance between two successive points. Here, the error made to measure the total thickness of a section is rather used to provide a minimum amount of thickness uncertainty. It will be indeed very hard to do better on short distances than what is done on a long, average distance.

**CZ: Lines 143, 223: Do I understand correct that you interpolate all time series (also with spacing of ∼0.2 m and ∼0.38 m) at 0.01 m intervals? Is this necessary and useful, and does this oversampling influence your results?**
» The authors: When linearly interpolating at the average sample step of the original series, we can reduce the amplitude of the high frequencies, independently of the error made on measuring the sample distance (Hinnov et al., 2002). So we overinterpolated at 0.01m to not create this bias in the analysis. However, we acknowledge that this

procedure results in an inflation of the AR-1 coefficient of the red-noise fit. In the revised version of the manuscript, we will linearly interpolate the series at the median sample distance, as also suggested by Linda Hinnov (the other referee). To limit the loss of power in the high frequencies, we designed an optimized interpolation scheme, that will be applied in the revised version of the manuscript. This optimized strategy will be based on the minimal average offset between the original sample positions and the interpolated sample positions.

**CZ: Lines 159-164: Your approach is good, but personally I would propose to also determine 95% confidence intervals of power by considering not only the average power spectrum from simulations. This may facilitate to compare (integrated) precession and obliquity power for paleoclimate studies.**
» The authors: This is a great idea! That will be applied in the next version of the manuscript

**CZ: 175ff: a table summarizing the results presented may be helpful in addition.**
» The authors: Another great idea to make the results clearer and present them in a concise form that will help the readers

**CZ: In Fig. 4 the confidence levels of the MTM and Lomb-Scargle spectra are different. I would propose to mention this in the figure caption.**
» The authors: That's true! We will explicitly mention that in the next version of the manuscript

**CZ: 10, 13: maybe express Nyquist frequency as sampling interval to be clearer**
» The authors: This is another good idea to make the things clearer! Knowing the fact that the Nyquist frequency is twice the sample step, it is very easy to convert the

percentage of the Nyquist frequency to number of sample steps. For instance, 20% of the Nyquist Frequency represents 10 times the sample step.

**CZ: 15-17: "In addition, the simulations indicate that taking at least 6-10 samples per precession cycle should allow calculation of robust power spectra estimates in the Milankovitch band." – This is not limited to precession I think, what about a more general statement as 'In addition, the simulations indicate that taking at least 6-10 samples per cycle should allow calculation of robust power spectra estimates in the respective cycle band'?**

» The authors: The reviewer is right. This requirement is actually valid for shortest cycle to be analysed, whatever its origin and period (obliquity, eccentricity or solar cycles).

**CZ: 28-29: "In core sediments, uncertainties in the sample position are also observed when performing physical sampling at very high resolution or because of core expansion phenomena (Hagelberg et al., 1995)" – suggestion: 'In cored sediments, uncertainties in the sample position are also observed when performing physical sampling at very high resolution or because of core expansion phenomena (Hagelberg et al., 1995) or imperfect coring (Ruddiman et al., 1987).'**

» The authors: We would like to thank the reviewer for this suggestion. It is indeed very important to say that core sections are not devoid of bias. We will rephrase as suggested.

**CZ: 37-38: "In this study, we address this problem by quantifying the impact of such errors on the frequency, as well as the power of higher-frequency cycles." ! the second part of this sentence ("the frequency, as well as the power of higher-frequency cycles") may be 'the frequency and power distributions'?**

» The authors: We thank the reviewer for this suggestion, which makes the sentence

much clear. We will rephrase as suggested.

**CZ: 42-44: This sentence seems in contradiction to the last sentence of the abstract, more consistent phrasing may solve this.**

» The authors: I think the reviewer refers to this sentence: "Based on our results, we suggest that one should take at least ∼10 measurements per high-frequency cycle in order to provide robust estimates of the power of the high-frequency cycles." And that is in contradiction with the last sentence of the abstract in which we said 6-10 samples per thinnest cycle targeted are necessary to identify all necessary cycles in the band we wish to explore. The authors apologize for this inconsistency and we will change "∼10 measurements" by "6-10 samples per highest-frequency cycle..." for more consistency with the abstract.

**CZ: 48: delete 'correctly'? 64: remove 'easily'**

» The authors: OK for both

**CZ: 98/99: could you mention that these are Devonian, and give a rough age as for the La Charce section?**

» The authors: OK for precising the ages of the sections. The ages of the La Thure section (Givetian, middle Devonian) are around 380 Ma (De Vleeschouwer and Parnell, 2014).

**CZ: 108: are the two brackets necessary?**

» The authors: Sorry for that misspelling. We will remove one of the brackets in the next version

**CZ: 119/120: "with an average of 110.3 ± 5.1 m, and a relative uncertainty of**

**4.6%" I would propose to mention that the "5.1 m" and "uncertainty of 4.6%" are estimated from only three experiments, and that these are regarded as representative, but may not be actually.**

» The authors: The comment from Linda Hinnov in page C2, bullet point 4, perfectly illustrates your comment: their team measured the La Charce section twice and found 112 m and 132 m thicknesses, either an average of 122 ± 10 m. The uncertainty is (10/122*100) 8.2% of the total thickness of the series. So our estimate is based on published data, but according to the personal comments from Linda Hinnov, available online in the second referee comments, this can be larger from a team to another.

**CZ: 146: maybe give also reference to the R package used ('dplR')**

» The authors: We will mention that Lomb-Scargle analyses have been done with the dplR in the next version.

**CZ: 155: "The confidence levels of the datasets were calculated before randomisation and directly plotted to the simulated spectra." I am unsure how this is meant, and I would suggest phrasing this more clearly.**

» The authors: The authors apologize for this unclear phrasing. That only means that we plotted in the 1,000 randomized spectra the AR1-confidence levels calculated in the original series to make easier the comparison of powers.

**CZ: 160-164: "Pori: the power spectrum before randomization" – as you calculate this for individual frequencies, following may be more clear: 'Pori: power before randomization for a specific frequency', same for Pave (if I understand this correct).**

» The authors: The reviewer is right. The rigorous phrasing should be: $P_{ori}(f)$: power before randomization at frequency f. The same for the others. We will rephrase in the next version of the manuscript

**CZ: 172 "with 5% uncertainty" – maybe clarify as 'with 5% stratigraphic uncertainty'**

» The authors: This change will be done

**CZ: 200: with "first frequency" 'lowest frequency' is meant I assume – could you clarify this?**

» The authors: The reviewer is right, and we now see that our phrasing was ambiguous because it depends if we read the spectrum from the left or from the right. The phrasing suggested by the reviewer should eliminate our ambiguous phrasing.

**CZ: 205-211: Please make clearer that geological data usually have no precise frequencies, but frequency ranges. You mention this, but I am not sure if everyone will understand this easily.**

» The authors: We suggest to mention after the sentence line 210: "For instance, because of variations of the sedimentation rates, the sedimentary expression of the orbital cycles is not focalised on specific frequencies but rather expressed on ranges of frequencies"

**CZ: 217: I am not sure if you need to mention "that the stratigraphic order of the samples in the raw series is preserved after randomisation" again. You develop this earlier in the manuscript.**

» The authors: We agree that this statement has been repeated and is superfluous in this line. We will delete this statement at 217 in the next version of the manuscript.

**CZ: 220f: "This difference realistically simulates small thickness errors, which accumulate when measuring successive sample steps." – this can in my opinion**

**be formulated better, and should highlight that errors may accumulate, or may also not accumulate but level out.-**

» The authors: We suggest the following sentence to rephrase: "this difference is interpreted as the simulation of small thickness measurement errors, which accumulate when measuring successive sample steps".

**CZ: 241: "above 40% of the Nyquist frequency", I would suggest to also mention the frequency, maybe in brackets after this statement. Maybe bring these ratios in direct reference to precession (e.g. _1/3rd of precession frequency/wavelength), so that this is more clear for readers not so familiar with time series analysis.**

» The authors: ok for both suggest, and we will also precise the equivalent in terms of number of sample steps, as suggested in a previous comment from the reviewer.

**CZ: 258/59: "As in the case of the La Charce series, the stratigraphic order of the samples is preserved in the randomised series" – In my opinion this is clear by now in the manuscript, and does not need to be repeated.**

» The authors: OK for removing this piece of information

**CZ: 304: replace "powers" by "power"**

» The authors: OK

**CZ: 310: "result suggest" – one of these need an "s" in the end**

» The authors: result needs an "s" in the end

**CZ: 312/13: "This requires that more than 6 samples per precession cycle have to be taken" - samples or measurements?**

» The authors: The authors see the ambiguity and regret it. We are talking about number of samples to take per precession cycles. This will be clarified in the manuscript

**CZ: 355: 356: maybe also refer to (Meyers, 2015; Shackleton et al., 1995)**
» The authors: OK for adding the references

**CZ: 396: "on the field" – in the field?**
» The authors: The correct expression is "in the field". Will be corrected in the next version

References :

Boulila, S., Galbrun, B., Miller, K.G., Pekar, S.F., Browning, J.V., Laskar, J., Wright, J.D., 2011. On the origin of Cenozoic and Mesozoic "third-order" eustatic sequences. Earth-Science Reviews 109, 94-112.

De Vleeschouwer, D., Parnell, A.C., 2014. Reducing time-scale uncertainty for the Devonian by integrating astrochronology and Bayesian statistics. Geology 42 (6), 491-494.

De Vleeschouwer, D., Boulvain, F., Da Silva, A.C., Labaye, C., Claeys, P., 2015. The astronomical calibration of the Givetian (Middle Devonian) timescale (Dinant Synclinorium, Belgium), from : Da Silva, A.-C., Whalen, M.T., Hladil, J., Chadimowa, L., Chen, D., Spassov, S., Boulvain, F., Devleeschouwer, X. (Eds.): Magnetic Susceptibility Application: A Window onto Ancient Environments and Climatic Variations. Geological

Society, London, Special Publications 414, 245 – 256.

Hinnov, L.A., Schulz, M., Yiou, P., 2002. Interhemispheric space–time attributes of the Dansgaard–Oeschger oscillations between 100 and 0 ka. Quaternary Science Reviews 21, 1213-1228.

Westerhold, T., Röhl, U., 2009. High resolution cyclostratigraphy of the early Eocene – new insights into the origin of the Cenozoic cooling trend. Climate of the Past 5, 309-327.

Zachos, J., Pagani, M., Sloan, L., Thomas, E., Billups, K., 2001. Trends, Rhythms, and Aberrations in Global Climate 65 Ma to Present. Science 292, 686-693.

---

## Author Response (AR1)

**Answers to comments from Christian Zeeden**

**Christian Zeeden (CZ): the La Thure series shows both precession and obliquity. Could you exemplary discuss what the result from your test means for this example record, and how it aids the interpretation?**

>> The authors: This example is indeed interesting because both obliquity and precession have been observed (De Vleeschouwer et al., 2015), with obliquity having higher powers than precession. Several studies suggested that a dominance of obliquity in tropical sediments reflect cooling or icehouse conditions, while dominance of precession would be associated to greenhouse conditions (Zachos et al., 2001; Westerhold and Röhl, 2009; Boulila et al., 2011). With the implementation of our test we show that precession nearly vanishes at 15% of uncertainty. As a result, if one does not take into account this sampling bias on power spectrum one can misleadingly interpret a dominance of obliquity in sediments, which impacts in turn on the climatic interpretations.

**CZ: explain what the Nyquist frequency represents.**

>> The authors: The Nyquist frequency is the highest frequency (or smallest period) that can be detected. It corresponds to the inverse of twice the sample step. This information will be added in the next version of the manuscript.

>> Done. A short definition of the Nyquist frequency is provided lines 224-225 (line numbers from the marked-up manuscript below).

**CZ: you suggest uncertainty to be fully random. I propose to briefly discuss why you assume this – and what effect(s) systematic uncertainty may have.**

>> The authors: We assume a fully random error, based on comparisons with actual data of sample distances repeatedly measured on the La Charce series. This comparison went as follows: In a first step, the thicknesses of the individual beds were measured and a lithologic log was drawn based on these measurements. In a second step, samples were taken from the studied section every 20 centimetres and the sample positions were indicated on the lithologic log. After this second step, we observed that the distances between two successive samples was not exactly 20 cm, but rather ranged from 10 cm to 30 cm, with an average of 19.7 cm and a standard deviation of 2.5 cm. This observation was made by comparing the expected stratigraphic position of the $n^{th}$ sample (n x 20 cm) with the stratigraphic position of the bed the sample comes from in the lithologic log. The mismatch in sample position between the lithologic log and the bed from which the sample was taken can be quantified for every sample within the studied stratigraphic interval. We evaluated the distribution of every sample's mismatch and observed a log-normal distribution. This observation is the basis for our suggestion to consider the stratigraphic uncertainty to be fully random.

In addition, the total thickness of the series was measured at 109,33 m. With this thickness we expected to take 547 samples. Instead, we took 555 samples. We thus have an error of 8 samples out of 555 samples, either 1.4% difference. This is of course much lower than the actual thickness measurement error for individual sample distances, which implies no systematic error.

We will briefly discuss the absence of systematic error in the revised manuscript

>> Done. Lines 94-102, we add a new condition for justifying the fact that no systematic error is made when measuring each sample distance independently from the previous measurements

**CZ: Lines 117-120: 106-116m is the overall spread in section thickness. From a conceptual point of view I think that this spread can hardly directly be used to estimate uncertainty in sample distance, because you see a result of ~550 (gamma distributed) sample distances summed up. Several of these will be shorter and longer than 20 cm – so your relative uncertainty will probably be higher – or fully systematic.**

>> The authors: The reviewer is right. On average, the error made to measure the entire section is lower than the error made to measure sample distances. On entire sections, systematic errors will have for consequence overestimate the thickness of certain parts of the section while other parts will be have underestimated thicknesses. Thus, the error made to measure the total thickness of the section will be lower than the distance between two successive points. Here, the error made to measure the total thickness of a section is rather used to provide a minimum amount of thickness uncertainty. It will be indeed very hard to do better on short distances than what is done on a long, average distance.

>> This answer does not need any change in the manuscript

**CZ: Lines 143, 223: Do I understand correct that you interpolate all time series (also with spacing of ~0.2 m and ~0.38 m) at 0.01 m intervals? Is this necessary and useful, and does this oversampling influence your results?**

>> The authors: When linearly interpolating at the average sample step of the original series, we can reduce the amplitude of the high frequencies, independently of the error made on measuring the sample distance (Hinnov et al., 2002). So we overinterpolated at 0.01m to not create this bias in the analysis. However, we acknowledge that this procedure results in an inflation of the AR-1 coefficient of the red-noise fit. In the revised version of the manuscript, we will linearly interpolate the series at the median sample distance, as also suggested by Linda Hinnov (the other referee). To limit the loss of power in the high frequencies, we designed an optimized interpolation scheme, that will be applied in the revised version of the manuscript. This optimized strategy will be based on the minimal average offset between the original sample positions and the interpolated sample positions.

>> Done. The optimized linear interpolation is now based of best-fit curve between the original time series and the time series that has been resampled at the mean sample distance of the original time series. Changes are now observed in the last four figures of the manuscript. An appendix has been added to detail the optimized linear interpolation, and the method has been updated in lines 227-232.

**CZ: Lines 159-164: Your approach is good, but personally I would propose to also determine 95% confidence intervals of power by considering not only the average power spectrum from simulations. This may facilitate to compare (integrated) precession and obliquity power for paleoclimate studies.**

>> The authors: This is a great idea! That will be applied in the next version of the manuscript

>> Done. The grey areas in the last four figures represent the 95% confidence intervals of power (Figs. 3 and 8-11).

**CZ: 175ff: a table summarizing the results presented may be helpful in addition.**

>> The authors: Another great idea to make the results clearer and present them in a concise form that will help the readers

>> Done. See Table 2

**CZ: In Fig. 4 the confidence levels of the MTM and Lomb-Scargle spectra are different. I would propose to mention this in the figure caption.**

>> The authors: That's true! We will explicitly mention that in the next version of the manuscript

>> Differences exist in the confidence levels between the MTM method and the Lomb-Scargle method due to different degrees of freedom between the two approaches. The $2\pi$-MTM analysis has a degree of freedom of ~6 (Mann and Lees, 1996), while the Lomb-Scargle method has a degree of freedom of 2 (Schulz and Stattegger, 1997). Differences are also observed in the La Thure series, which exhibits a high autocorrelation coefficient value. Addition al information is added

**CZ: 10, 13: maybe express Nyquist frequency as sampling interval to be clearer**

>> The authors: This is another good idea to make the things clearer! Knowing the fact that the Nyquist frequency is twice the sample step, it is very easy to convert the percentage of the Nyquist frequency to number of sample steps. For instance, 20% of the Nyquist Frequency represents 10 times the sample step.

>> Done. See notably Figs. 3, 8-11 and Table 2

**CZ: 15-17: "In addition, the simulations indicate that taking at least 6-10 samples per precession cycle should allow calculation of robust power spectra estimates in the Milankovitch band." – This is not limited to precession I think, what about a more general statement as 'In addition, the simulations indicate that taking at least 6-10 samples per cycle should allow calculation of robust power spectra estimates in the respective cycle band'?**

>> The authors: The reviewer is right. This requirement is actually valid for shortest cycle to be analysed, whatever its origin and period (obliquity, eccentricity or solar cycles).

>> In the manuscript, we replaced 4-10 samples per precession to 4-10 samples per thinnest cycles of interest. The case of the precession is maintained as example for the case of studies focused on the Milankovitch band (e.g., lines 30, 34, 517, 543 in the abstract and conclusion).

**CZ: 28-29: "In core sediments, uncertainties in the sample position are also observed when performing physical sampling at very high resolution or because of core expansion phenomena (Hagelberg et al., 1995)" – suggestion:**

'In cored sediments, uncertainties in the sample position are also observed when performing physical sampling at very high resolution or because of core expansion phenomena (Hagelberg et al., 1995) or imperfect coring (Ruddiman et al., 1987).'

>> The authors: We would like to thank the reviewer for this suggestion. It is indeed very important to say that core sections are not devoid of bias. We will rephrase as suggested.

>> Done (line 58).

**CZ: 37-38: "In this study, we address this problem by quantifying the impact of such errors on the frequency, as well as the power of higher-frequency cycles." ! the second part of this sentence ("the frequency, as well as the power of higher-frequency cycles") may be 'the frequency and power distributions'?**

>> The authors: We thank the reviewer for this suggestion, which makes the sentence much clear. We will rephrase as suggested.

>> Done (lines 69-70)

**CZ: 42-44: This sentence seems in contradiction to the last sentence of the abstract, more consistent phrasing may solve this.**

>> The authors: I think the reviewer refers to this sentence: "Based on our results, we suggest that one should take at least ~10 measurements per high-frequency cycle in order to provide robust estimates of the power of the high-frequency cycles." And that is in contradiction with the last sentence of the abstract in which we said 6-10 samples per thinnest cycle targeted are necessary to identify all necessary cycles in the band we wish to explore. The authors apologize for this inconsistency and we will change "~10 measurements" by "6-10 samples per highest-frequency cycle…" for more consistency with the abstract.

>> This answer does not need any change in the manuscript

**CZ: 48: delete 'correctly'?**
**64: remove 'easily'**
>> The authors: OK for both

>>Done

**CZ: 98/99: could you mention that these are Devonian, and give a rough age as for the La Charce section?**
>> The authors: OK for precising the ages of the sections. The ages of the La Thure section (Givetian, middle Devonian) are around 380 Ma (De Vleeschouwer and Parnell, 2014).

>> Done (lines 171-172)

**CZ: 108: are the two brackets necessary?**
>> The authors: Sorry for that misspelling. We will remove one of the brackets in the next version

>> Done (line 188)

**CZ: 119/120: "with an average of 110.3 ± 5.1 m, and a relative uncertainty of 4.6%" I would propose to mention that the "5.1 m" and "uncertainty of 4.6%" are estimated from only three experiments, and that these are regarded as representative, but may not be actually.**
>> The authors: The comment from Linda Hinnov in page C2, bullet point 4, perfectly illustrates your comment: their team measured the La Charce section twice and found 112 m and 132 m thicknesses, either an average of 122 ± 10 m. The uncertainty is (10/122*100) 8.2% of the total thickness of the series. So our estimate is based on published data, but according to the personal comments from Linda Hinnov, available online in the second referee comments, this can be larger from a team to another.

**CZ: 146: maybe give also reference to the R package used ('dplR')**
>> The authors: We will mention that Lomb-Scargle analyses have been done with the dplR in the next version.

>> Done (lines 238-239)

**CZ: 155: "The confidence levels of the datasets were calculated before randomisation and directly plotted to the simulated spectra." I am unsure how this is meant, and I would suggest phrasing this more clearly.**
>> The authors: The authors apologize for this unclear phrasing. That only means that we plotted in the 1,000 randomized spectra the AR1-confidence levels calculated in the original series to make easier the comparison of powers.

>> A comment from Linda Hinnov requested to calculate the confidence levels for each simulation, so this part has been updated to take into account the comment from the other reviewer (lines 252-253)

**CZ: 160-164: "Pori: the power spectrum before randomization" – as you calculate this for individual frequencies, following may be more clear: 'Pori: power before randomization for a specific frequency', same for Pave (if I understand this correct).**
>> The authors: The reviewer is right. The rigorous phrasing should be: Pori($f$): power before randomization at frequency $f$. The same for the others. We will rephrase in the next version of the manuscript

>> Done (lines 264-265)

**CZ: 172 "with 5% uncertainty" – maybe clarify as 'with 5% stratigraphic uncertainty'**
>> The authors: This change will be done

>> Done (line 276)

**CZ: 200: with "first frequency" 'lowest frequency' is meant I assume – could you clarify this?**
>> The authors: The reviewer is right, and we now see that our phrasing was ambiguous because it depends if we read the spectrum from the left or from the right. The phrasing suggested by the reviewer should eliminate our ambiguous phrasing.

>> Done (throughout sections 6 and 7)

**CZ: 205-211: Please make clearer that geological data usually have no precise frequencies, but frequency ranges. You mention this, but I am not sure if everyone will understand this easily.**
>> The authors: We suggest to mention after the sentence line 210: "For instance, because of variations of the sedimentation rates, the sedimentary expression of the orbital cycles is not focalised on specific frequencies but rather expressed on ranges of frequencies"

>> Done (lines 327-329)

**CZ: 217: I am not sure if you need to mention "that the stratigraphic order of the samples in the raw series is preserved after randomisation" again. You develop this earlier in the manuscript.**
>> The authors: We agree that this statement has been repeated and is superfluous in this line. We will delete this statement at 217 in the next version of the manuscript.

>> Done

**CZ: 220f: "This difference realistically simulates small thickness errors, which accumulate when measuring successive sample steps." – this can in my opinion be formulated better, and should highlight that errors may accumulate, or may also not accumulate but level out.-**
>> The authors: We suggest the following sentence to rephrase: "this difference is interpreted as the simulation of small thickness measurement errors, which accumulate when measuring successive sample steps".

>> Done. We finally do not need to add this piece of information since we precised that error model in random and not systematic (lines 94-102).

**CZ: 241: "above 40% of the Nyquist frequency", I would suggest to also mention the frequency, maybe in brackets after this statement. Maybe bring these ratios in direct reference to precession (e.g. _1/3rd of precession frequency/wavelength), so that this is more clear for readers not so familiar with time series analysis.**
>> The authors: ok for both suggest, and we will also precise the equivalent in terms of number of sample steps, as suggested in a previous comment from the reviewer.

**CZ: 258/59: "As in the case of the La Charce series, the stratigraphic order of the samples is preserved in the randomised series" – In my opinion this is clear by now in the manuscript, and does not need to be repeated.**

>> The authors: OK for removing this piece of information

>> Done

**CZ: 304: replace "powers" by "power"**
>> The authors: OK

>> Done

**CZ: 310: "result suggest" – one of these need an "s" in the end**
>> The authors: result needs an "s" in the end

>> Done

**CZ: 312/13: "This requires that more than 6 samples per precession cycle have to be taken" - samples or measurements?**
>> The authors: The authors see the ambiguity and regret it. We are talking about number of samples to take per precession cycles. This will be clarified in the manuscript

**CZ: 355: 356: maybe also refer to (Meyers, 2015; Shackleton et al., 1995)**
>> The authors: OK for adding the references

>> Done (line 598)

**CZ: 396: "on the field" – in the field?**
>> The authors: The correct expression is "in the field". Will be corrected in the next version

>> Done (lines 585, 632)

**Linda Hinnov (LA): The authors call on the gamma probability density distribution to characterize stratigraphic sampling. Here there could be more explanation, e.g., a simple illustration of the problem, i.e., in Figure 1 add a diagram of a hypothetical stratigraphic section, different sampling sequences, and their histograms – perhaps the same ones as presented in Figure 2);**

>> The authors: We thank the reviewer for this interesting suggestion that will help the reader to understand the problem. We actually have prepared a figure to illustrate the problem showing a hypothetical series with positions of samples obviously non-equally spaced. The diagrams used in real examples will be reused here, as suggested

>> Done: new Figure 1 illustrates the problem

**LA: in Figure 1 caption indicate "gampdf(x, k, θ)" and label horizontal axis as "x". The models presented in Figure 2 displayed in F, G and H: what values of k and θ do these correspond to?**

>> The authors: $k$ and $\theta$ can be easily calculated using equations (3) and (4) of the manuscript. The mean sample distance is 1 unit in this case, and we performed the gamma test using setting the standard deviation at 0.05, 010 and 0.15 units respectively. In the 3 cases, k and θ values are as follows:

- Sd=0.05 units: θ=0.0025 and k=400
- Sd=0.10 units: θ=0.01 and k=100
- Sd=0.15 units: θ=0.0225 and k=44

This piece of information will be added in the revised version of the manuscript

>> Done (new Figure 3)

**4. Implementation of the models in the stratigraphic-uncertainty tests**

**LA: This reviewer can personally attest to the difficulty in measuring a consistent thickness for the same outcrop by different researchers - in my experience in one case: 112 m vs. 132 m! For overturned sections, any dip error committed will contribute to a positive bias in stratigraphic thickness measurements. There is undoubtedly such a problem in the steeply dipping Cretaceous section at La Charce examined in this paper.**

>> The authors: We thank the reviewer for this comment that was reused in the answer to referee 1. This example supports our idea that reaching a constant sample step on geological data is not trivial.

>> This discussion does not need revision in the manuscript

**LA: On issues concerning methods, it is important to restrict interpolation to mean or median rate when applying AR noise models with MTM spectra (such as used in SSA-MTM Toolkit). The Devonian section has a mean sample rate of 0.38 m – not clear what the median rate is – and this is much larger than the interpolation to 0.01 m. The Cretaceous section has a mean sample rate of 0.20 m, so has a similar problem. The authors should recalculate the MTM analysis with interpolation to the median sample spacing of the two sections. (The red noise spectra will be significantly different because of the way the autocorrelation lag-1 coefficient is calculated.) The other parameter that requires reporting is whether "log" or "linear" fitting was enabled in the calculation of robust red noise for the MTM spectra.**

>> The authors: The other reviewer (Christian Zeeden) has also commented on the overinterpolation procedure. Basically, we will provide a new method of interpolation, in order to optimize this step and limiting the loss of power in the high frequencies, that naturally occurs when resampling at the mean sample distance (see Hinnov et al., 2003).

>> Correction done, interpolation is now optimized in order to limit the loss of powers in the high frequencies. Data are linearly interpolated at the average sample distance of the original dataset (lines 227-232 + Appendix A)

As for the comment on the linear or log-fit, we employed a linear fit, from Meyers' astrochron ML96 function. In this function, the method for calculating the background median smooth fit has been modified by entering a Tukey's robust end point rule for the very low frequencies, which allows the level of lag-1 coefficient to be increased. This is below what the help of mtm.ML96 function says:

*"This function conducts the Mann and Lees (1996; ML96) "robust red noise" analysis, with an improved median smoothing approach. The original Mann and Lees (1996) approach applies a truncation of the median smoothing window to include fewer frequencies near the edges of the spectrum; while truncation is required, its implementation in the original method often results in an "edge effect" that can produce excess false positive rates at low frequencies, commonly within the eccentricity-band (Meyers, 2012).*

*To help address this issue, an alternative median smoothing approach is applied that implements Tukey's robust end-point rule and symmetrical medians (see the function runmed for details)."*

>> The median-fit of the red-noise background of La Thure was previously based on a "linear" comparison of powers. We now used a comparison of log-powers, much consistent with the red-noise background of La Thure with the REDFIT analysis (new Figures 6c, 10).

**5. Application to a sum of sinusoids**

**LA: This section quantifies the loss of power at high frequencies with increasing uncertainty of (variability in) the sample step sequence for a simulated sum-of-sinusoids series. The absence of windowing in the Lomb-Scargle (LS) spectra would be expected to result in higher spectral variance compared to multitaper-windowed MTM spectra, and may account for the elevated grey spectra from the LS Monte Carlo simulations (compared to those of the MTM spectra). Interestingly, for 10% and 15% σ, loss of power occurs at practically the same frequencies in both MTM and LS spectra. Would it be possible to indicate the expected variance in Nyquist frequency for the 3 cases (5%, 10%, 15%) in order to understand the accuracy of the MTM and LS spectra? A new order of the graphs in Figures 2 and 3 might benefit the presentation:**

- **New Figure 2: display Figs. 2F, G, H only, and explain how these relate to k and θ (or put them into a Figure 1B).**

- **New Figure 3: in top row, display Fig. 2A, B, C, D; bottom row display Fig. 3A, B, C, D.**

- **Figs. 2E and 3E could be placed into a new figure.**

>> The authors: We will modify the figures as suggested and ask the reviewer to further clarification about the variance question.
>> Figures modified (new Figures 3 and 4)
>> As for the variance question, the reviewer is right; the grey-spectra level is much elevated in the Lomb-Scargle analysis than in the MTM analysis. Following the recommendation of Christian Zeeden, we now display in Figure 3 and from Figures 8 to 11 the 95% zone of the 1000 simulations, which we assume to represent the 2σ-interval of the power spectrum estimates. We think than displaying the curve is much meaningful than the value of variance at the Nyquist frequency.

**LA: What did we learn from this exercise and how will it help with the interpretation of the geological datasets to follow?**

>> The authors: This exercise is performed on a pure sinusoid signal, not related to any geological data, and having an arbitrary sample step. It shows the general pattern of disturbing the sampling interval on the power spectrum, independently of the nature of the geological data (finite length, noisy and non-strictly periodic). In this case, the power spectrum can be controlled and be fixed as equal for all spectral peaks, which helps to examine the relative change in power throughout the spectrum.

>> In the sum of sinusoid case, we now emphasise the loss of power in the high frequencies, since the power spectrum of spectral peak can be controlled. In the real geological example, we rather emphasise on the loss of significance level in the high frequencies, which is a direct consequence of the loss of power in the high frequencies, and which has the most implications for matching sedimentary series to insolation series.

**6. Application to geological datasets**

**LA: The MTM spectrum of the Devonian series (Figure 4D) shows a robust red noise model with extremely elevated low frequencies, implying that a "log" fit was calculated in SSAMTM Toolkit, and that the model suffers additionally from the 0.01 m interpolation (see comments for Section 4). Some of the text in this section about differences in red noise calculations (which by the way are not meaningfully explained) may not be needed once the interpolation problem is addressed.**

>> The authors: To calculate the spectrum of the La Thure section with the confidence levels, we used the mtm.ML96 function from astrochron package. A linear model of background fit was used (please find below the code line we applied:

```
ML96_1  = mtmML96(dat_pad1,tbw=2,ntap=3,padfac=1,demean=T,detrend=T,medsmooth=0.2,
          opt=3,linLog=1,siglevel=0.95,output=1,CLpwr=T,xmin=0,xmax=1/(2*dtmoy),
          sigID=F,pl=2,genplot=F,verbose=F)
```

linLog=1 means we used a linear fit model.

Definitely, we will re-fix the resample step at the median sample distance, which is 0.30 m.

>> The spectral background of the La Thure section is now calculated based on a comparison of log power, instead of linear powers previously. The results are much consistent with the red-noise fit calculated with the REDFIT method.
>> We decided to fix the sample step at 0.38 m for La Thure, which corresponds to the average sample distance of the series. This choice is motivated by the fact that only 30% of the series is sampled at a density equal or higher than the median sample distance, while 43% of the series is sampled at a density equal or higher than the average sample distance.

**7. Discussion**

**LA: The main point of this study is that sampling is the critical decision that must be made when evaluating a stratigraphic sequence for paleoclimate signals. Almost all problems can be controlled with high-density sampling, e.g., 6-10 samples per putative precession cycle. It appears that one can easily expect 5% errors in stratigraphic position measurements, which combined with sedimentation rate variations, will mix the highest frequencies of a sampled sequence. Thus we are always alarmed at how low in power – and misaligned – precession cycles are in stratigraphic spectra. In the end, one never knows if a sample that has been collected has been assigned to its true stratigraphic position. This is an important limitation that is under-appreciated by the geological community and the authors should be commended for tackling this problem.**

>> The authors: We thank the reviewer for this very positive comment, which will probably feed the discussion of the revised version of the manuscript.
>> This was implemented in the discussion from lines 601-624.

**LA: A number of issues have been left unexplored: (1) how does systematic sample position error, such as can occur with receded marls alternating with prominent limestones in outcrops, affect stratigraphic spectra; (2) can astronomical tuning bypass the positional uncertainty problem (notwithstanding the recent approach described in Zeeden et al., 2015); and (3) how does the positional uncertainty problem affect the red noise model estimates?**

>> The authors: For question (1), we have shown to the other reviewer that the error is not systematic but fully random, even in the case of alternating sedimentation.

>> For question (2), the error in the sample position acts on average like variations of the sedimentation rate: it decreases the power spectrum in the high frequencies, and distributes the power of the obliquity and precession over a large range of frequencies. The approach of Zeeden et al (2015) applies a very wide bandpass filter which should limit the effect of such error, because a large of frequencies are taken into account in the filter. However, we acknowledge that in spectra of sedimentary series, it is common to observe a band of frequencies between the obliquity and precession for which we don't know if they are related to one or the other cycle. A combination of methods involving wide filters and evolutive spectral analysis should help in resolving this issue.

>> For question (3), this is an interesting question, and actually at this point, we do not have the answer to this question. However, this could be the topic of a follow-up study.

>> For question (3), we have included in the new manuscript the calculations of the red-noise confidence levels for each simulation. After 1000 simulations, the red-noise confidence levels are on average very similar to the confidence levels before randomisation and show very narrow dispersion. This stability is probably due to the fact that the sample distance randomisation implies a dispersion of the power spectrum on a broad spectral band. The fit of the spectral background being calculated on a broad band, the randomisation procedure does not change the average power calculated on a broad band. Implemented in Figs. 8-11, Table 1, and discussed from lines 601-608.

**Other comments**

**LA: Lines 23-24: The Multi-Taper Method (Thomson, 1982) might be more accurately characterized as a spectrum estimator that is based on the Fourier Transform – not as a derivative of the Fourier Transform.**
>> The authors: The reviewer is right. The multi-taper method is roughly the average of Fourier Transforms of the series studied weighted by windows called Slepian sequences.

**LA: Line 27: A recent massive improvement to the Jacob's staff in outcrop studies is terrestrial laser scanning with precision positioning at the mm level (Franceschi et al., 2011; Franceschi et al., 2015).**
>> The authors: We thank the reviewer for having provided us with this reference.

**LA: Line 101: Change to "Pas et al., 2015".**
>> The authors: OK

**LA: Lines 264- 265: what does the output of the "long-term trend of the variance" look like, and what was used to compute the "LOWESS regression with a 10% coefficient"?**
>> The authors: The figure of the detrend procedure will be added in the next version of the manuscript, and

**LA: Line 350: For monotonous stratigraphy yielding Milankovitch signal see also Latta et al., 2006.**
>> The authors: This reference is elder than the one we have cited in the original manuscript. OK to add this citation.
>> Done (line 639).

**LA: Line 355: Change "require" to "requires"**
>> The authors: OK

**LA: Line 361: Delete "Note than".**
>> The authors: OK

**LA: Supplementary File: R package dplR appears to be used but not referenced in the main text. R is used to calculate REDFIT– is it provided in the dplR package?**
>> The authors: The authors are really sorry for having forgotten to cite dplR package. The other referee, Christian Zeeden, has also noticed that. As we cited the astrochron package from Stephen Meyers, we also have to cite dplR package, which will be done in the revised version of the manuscript.
>> Done (lines 238-239).

[revised manuscript text omitted]

$E = k * \Theta$        (1)

and its variance ($\sigma^2$) as:

$\sigma^2 = k * \Theta^2 = E * \Theta$    (2)

Both the mean (*E*) and the variance ($\sigma^2$) are known, as they correspond to the mean and variance of the sample steps, and they can be quantified in the field (see Section 4

for a discussion on the variance of sample steps). Therefore, *k* and $\Theta$ can be parameterized using the following relations:

$\Theta = \dfrac{\sigma^2}{E}$        (3)

$k = \dfrac{E}{\Theta}$        (4)

Various versions of gamma probability density functions are shown in Fig. 2. A high variance-to-mean ratio corresponds to a high $\Theta$-parameter value compared to the value of the *k*-parameter. The resulting density probability function corresponds to an exponential probability function in the most severe and spectrum-destructive case.

This distribution corresponds to sampling conditions during which no control was exerted on the stratigraphic position of samples, so that the uncertainty on the sample position is at a maximum. Obviously, this situation is not a realistic case to reflect geologic practice.

In the opposite case, a low variance-to-average ratio corresponds to a low $\Theta$- parameter value compared to the value of the $k$-parameter. The resulting density probability function is close to a Gaussian curve, although bound on one side to 0, so that the curve has a positive support. This case corresponds to geologic sampling during which the position of each sample was carefully measured and reported with respect to the stratigraphic column. Nevertheless, even in this case, stratigraphic uncertainties exist, mainly because of outcrop or core conditions. Interestingly, this latter case has a similar distribution to the distribution of sample distances in the La

Charce series (Fig. 1e). This illustrates that the gamma model is well adapted for simulating the errors made on the measurement of the sample distances.

**3. The geological datasets**

Two geological datasets from previously published papers were used here to assess the effect of stratigraphic uncertainty on power spectra.

3.1. Gamma-ray spectrometry from La Charce (Valanginian, Early Cretaceous)

A total of 555 gamma-ray spectrometry measurements were performed *in situ* on the

La Charce section (Department of Drôme, SE France; Martinez et al., 2013, 2015).

The section is composed of marl-limestone alternations that were deposited in a hemipelagic environment during the Valanginian and Hauterivian stages (~134-132

Ma ago, Early Cretaceous; Martinez et al., 2015). Detailed analyses of their clay mineralogical, geochemical, faunal contents allowed these alternations to be attributed to orbital climate forcing. Gamma-ray spectrometry measurements have been used to discriminate 
[revised manuscript text omitted]

| | | σ=0% | σ=5% | σ=10% | σ=15% |
|---|---|---|---|---|---|
| **La Charce - MTM** | Autoregressive coefficient | 0.440 | 0.433 ± 0.025 | 0.432 ± 0.037 | 0.434 ± 0.048 |
| | Average power (x10⁻⁴) | 3.54 | 3.55 ± 0.13 | 3.58 ± 0.20 | 3.61 ± 0.25 |
| **La Charce - redfit** | Autoregressive coefficient | 0.468 | 0.468 ± 0.002 | 0.467 ± 0.003 | 0.467 ± 0.006 |
| | Average power | 0.398 | 0.399 ± 0.003 | 0.402 ± 0.005 | 0.407 ± 0.008 |
| **La Thure - MTM** | Autoregressive coefficient | 0.657 | 0.658 ± 0.025 | 0.653 ± 0.029 | 0.651 ± 0.033 |
| | Average power (x10⁻³) | 1.67 | 1.67 ± 0.04 | 1.67 ± 0.05 | 1.68 ± 0.07 |
| **La Thure - redfit** | Autoregressive coefficient | 0.407 | 0.406 ± 0.004 | 0.405 ± 0.008 | 0.404 ± 0.013 |
| | Average power | 0.890 | 0.894 ± 0.011 | 0.900 ± 0.019 | 0.904 ± 0.027 |

**Table 1.** Results of red-noise background estimates from the La Charce and the La Thure
series with the 2π-MTM and the REDFIT analyses.

| | | Level of stratigraphic uncertainty | | |
|---|---|---|---|---|
| | | **5%** | **10%** | **15%** |
| **La Charce MTM** | Highest frequency before smoothing
equivalent number sample steps | 81% Nyquist
2.5x | 58% Nyquist
3.4x | 43% Nyquist
4.7x |
| | Highest frequency confounded spectra
equivalent number sample steps | 27% Nyquist
7.4x | 19% Nyquist
10.8x | 18% Nyquist
11.3x |
| **La Charce REDFIT** | Highest frequency before smoothing
equivalent number sample steps | /
/ | 58% Nyquist
3.4x | 42% Nyquist
4.8x |
| | Highest frequency confounded spectra
equivalent number sample steps | 28% Nyquist
6.8x | 18% Nyquist
10.9x | 18% Nyquist
10.9x |
| **La Thure MTM** | Highest frequency before smoothing
equivalent number sample steps | 83% Nyquist
2.4x | 66% Nyquist
3.0x | 52% Nyquist
3.9x |
| | Highest frequency confounded spectra
equivalent number sample steps | 52% Nyquist
3.9x | 20% Nyquist
10x | 20% Nyquist
10x |
| **La Thure REDFIT** | Highest frequency before smoothing
equivalent number sample steps | /
/ | 53% Nyquist
3.8x | 53% Nyquist
3.8x |
| | Highest frequency confounded spectra
equivalent number sample steps | 52% Nyquist
3.9x | 22% Nyquist
9.3x | 20% Nyquist
10x% |

**Table 2.** Synthesis of the results of highest frequencies before smoothing of the spectra
when applying the Monte-Carlo simulations, and of highest frequency in which the spectra
before and after simulation can be confounded.